

# What caused record-breaking aerosol loading over the South China Sea in April 2023

**Saginela Ravindra Babu[1*] and Neng-Huei Lin[1,2*]**

[1]Department of Atmospheric Sciences, National Central University, Taoyuan 32001, Taiwan.

[2]Center for Environmental Monitoring and Technology, National Central University, Taoyuan 32001, Taiwan.

Correspondence to**:** S. Ravindra Babu (baburavindra595@gmail.com) and Neng-Huei Lin (nhlin@cc.ncu.edu.tw).

**Abstract:**

In April 2023, the South China Sea (SCS) experienced an unprecedented surge in aerosol loading, reaching the highest levels recorded in the two-decade Moderate Resolution Imaging Spectroradiometer (MODIS) satellite data period (2003–2023). Satellite observations revealed a 150% increase in aerosol optical depth (AOD) from MODIS and a 50% rise in carbon monoxide (CO) at 700 and 500 hPa from Measurements Of Pollution In The Troposphere (MOPITT) over SCS. Here, we investigate the drivers and atmospheric mechanisms responsible for this extreme event, identifying large-scale biomass burning (BB) across northern Peninsular Southeast Asia (PSEA), particularly Laos and Myanmar as the primary source. Our analysis indicates that anomalously high surface temperatures, low soil moisture, reduced precipitation, and a persistent upper-tropospheric anticyclone created favorable BB conditions over PSEA. Laos alone accounted for ~56% of the BB activity in the region, recording its largest monthly burned area (1.08 million hectares) since 2002. Dynamical analysis of the large-scale atmospheric circulation patterns revealed a major shift in regional wind regimes: the climatological south-westerlies over the SCS were replaced by anomalous northerlies, driven by the eastward shift of the Bay of Bengal anticyclone and the development of a cyclone anomaly over the western North Pacific (WNP). These changes redirected smoke transport from the usual WNP pathway to the SCS, resulting in significant transboundary pollution. This study highlights the critical role of compound meteorological extremes and circulation anomalies in amplifying regional aerosol loading, with implications for air quality, climate feedbacks, and environmental monitoring across Southeast Asia.

**Key words: Aerosol loading; South China Sea; MODIS; Wildfires**



**1. Introduction**
In the changing climate scenario, both natural and anthropogenic activities have contributed to a
continuous increase in surface temperatures worldwide over the past decade (Seneviratne et al.,
2021). In 2023, a record-high global mean surface temperature was observed, marking the warmest
period in the last seven months (June to December), surpassing the previous record set in 2016 by
a significant margin of 0.13°C to 0.17°C (Esper et al., 2024; Forster et al., 2024; Min, 2024;
Raghuraman et al., 2024). The extreme temperatures contributed to record-breaking wildfires
worldwide in 2023, with 70% of the total burning occurring in the Northern Hemisphere (Kolden
et al., 2024). Among all, Canadian wildfires emerged as the primary hotspot in 2023, with
significant fires in both the eastern and western regions causing notable increases in carbon
monoxide (CO) and tropospheric aerosols over the past twenty years (Liu et al., 2024). The
unprecedented wildfire season in Canada from May to September 2023 burned three times more
biomass than the previous record, leading to the highest annual carbon emissions from biomass
burning (BB) since 2015 (Byrne et al., 2024; MacCarthy et al., 2024). Furthermore, catastrophic
wildfires have also occurred in regions such as Hawaii, the Mediterranean, central Amazonia, and
central Chile (Roy et al., 2024; Lemus- Canovas et al., 2024; Espinoza et al., 2024; Jones et al.,
2024; Cordero et al., 2024). Greece experienced its most severe wildfire on record, with a burned
area of 96,000 hectares in 2023 (Michailidis et al., 2024). The August 2023 wildfires on Maui,
Hawaii, were among the deadliest U.S. wildfire incidents, resulting in 100 deaths and an estimated
loss of $5.5 billion (NOAA NCE, 2023). As a result of multiple fire spots across the globe, the
global mean concentrations of atmospheric carbon dioxide ($CO_2$), methane ($CH_4$), and nitrous
oxide ($N_2O$) reached new annual record highs of 419.3 ppm, 1922.6 ppb, and 336.7 ppb,
respectively. The global atmospheric $CO_2$ growth rate in 2023 was 2.79 ± 0.08 ppm (Ke et al.,
2024; Gui et al., 2024), the third-largest since 2000 and the fourth-largest since 1959.
The Asian region frequently experiences forest fires and biomass-burning activities,
significantly impacting the global carbon footprint (Xia et al., 2025). The South China Sea (SCS)
in Asia is the largest marginal ocean region in the tropical–subtropical western North Pacific. It is
a prime example of a marine area with minimal air pollution (Pani et al., 2023). This is further
supported by long-term satellite-measured Aerosol Optical Depth (AOD) spatial distribution. **Sup.**
**Figures 1a** and **1b** indicate shallow AOD values with minimal standard deviations in the SCS,



illustrating a clean marine environment. The SCS has a monsoon climate characterized by a northeast monsoon during winter and spring and a southwest monsoon during summer and autumn (Cui et al., 2016). These monsoon circulations allow natural and anthropogenic pollutants from East Asia to be lifted and transported over long distances to the SCS (Lin et al., 2013). During the summer monsoon season (August and September), the frequent burning of peat forests on the Maritime Continent (MC) also affects the adjacent regions of the southern SCS. In addition to East Asia and the MC, the springtime open biomass burning (BB) over Peninsular Southeast Asia (PSEA, including Myanmar, Thailand, Cambodia, Laos, and Vietnam) also impacts the SCS. The PSEA is one of the hotspot regions with the most intensive biomass-burning activities in the world (Lin et al., 2013; Reid et al., 2013) and is a major contributor to carbon emissions and atmospheric aerosols during the springtime (March-April). Open BB occurs almost every year during spring in the PSEA due to slash-and-burn agricultural activities (Lee et al., 2016; Tsay et al., 2016; Huang et al., 2020), emitting a substantial number of aerosols and trace gases into the atmosphere (Ou-Yang et al., 2022). The influence of aerosol loading over the SCS is strongly associated with the sources of aerosols and the prevailing wind circulation.

Although the unprecedented Canadian wildfires in 2023 garnered immense scientific interest and were well-documented in several studies, the record-breaking aerosol loading in the SCS in April 2023 received relatively little international attention. The historic event over the SCS in April 2023 can be observed from the Moderate Resolution Imaging Spectroradiometer (MODIS) Aqua AOD anomalies compared to the long-term mean (2003-2022), which shows extreme positive anomalies over the SCS and surrounding regions in April 2023, in contrast to the rest of the globe (**Fig. 1**). However, AOD anomalies in May further illustrate the absence of positive anomalies over the SCS and the presence of higher positive anomalies specifically over North America, particularly Canada. The time series of monthly mean AOD over the SCS further indicates a record-high AOD in April 2023 compared to the rest of the MODIS data from 2003 to 2023 (**Sup. Fig. 2d**). The exceptional record-breaking aerosol loading in April 2023 is unusual for remote marine locations such as the SCS and requires detailed investigation. In this communication, we examine the factors and physical processes that contributed to the unprecedented aerosol levels observed in April 2023, utilizing extensive data collected from multiple sources over an extended period. The following three major topics are explored in detail in the present study:



- How extreme are these AOD/CO anomalies, and what magnitude was increased?
- What are the sources for these record-breaking aerosol loadings over SCS?
- Were dynamic and large-scale circulations responsible for this event?

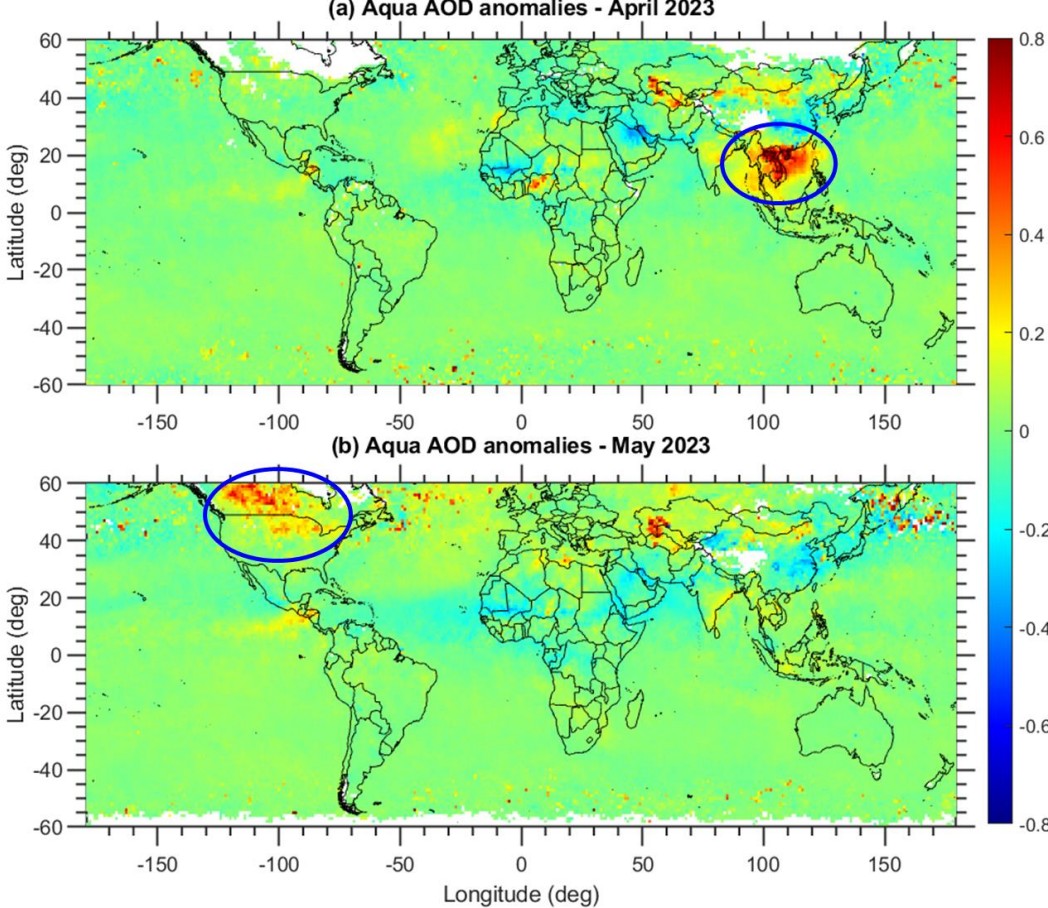

**Figure 1.** MODIS Aqua measured AOD anomalies in (a) April 2023 and (b) May 2023 compared to the long-term mean (2003-2022). The highlighted circles in (a) and (b) indicate the AOD anomalies over the South China Sea (SCS) and Canada regions. This figure highlights that the AOD anomalies observed by MODIS are significant and particularly pronounced over the SCS compared to the other areas globally. It illustrates the unique characteristics of April 2023 in terms of climatology. Data visualizations produced using MATLAB 2023b (https://matlab.mathworks.com).



**2. Data and Methodology**
**2.1.1 Data**
This study relies entirely on publicly available data, covering the period from 2003 to 2023. We
used data products from various satellite measurements. For example, AOD data is obtained from
MODIS, while CO data is obtained from MOPITT and AIRS satellites, respectively. The
tropospheric column ozone data are obtained from OMI/MLS along with AIRS ozone data at 700
hPa and 500 hPa, respectively.
**Moderate Resolution Imaging Spectroradiometer (MODIS)**
MODIS is a passive sensor aboard the Aqua and Terra satellites, which are in a sun-synchronous
orbit, and pass the Equator in the morning (Aqua) and afternoon (Terra). From MODIS satellite
measurements, we utilized aerosol optical depth (AOD), fire counts, fire radiative power (FRP),
cloud fraction, and burned area products. We used Level 3 monthly AOD at $1\circ \times 1\circ$ spatial
resolution derived from the mean of the Dark Target and Deep Blue Combined Aerosol Products
from the Terra satellite (MOD08_M3 Collection 6.1) and Aqua satellite (MYD08_M3 Collection
6.1) (Platnick et al., 2015; Buchholz et al., 2020). Additionally, we utilized MODIS's product of
daily fire counts and fire radiative power (FRP) (Giglio et al., 2006, 2016, 2018). Direct fire counts
from MODIS were obtained from the Fire Information for Resource Management System
(FIRMS) dataset. We selected all MODIS fire counts from the Terra and Aqua sensors with a
confidence level of 80% or higher. Each month, the total MODIS daily fire counts and FRP are
constructed and gridded at a resolution of 0.25° latitude × 0.25° longitude. Finally, we utilized Cloud
Fraction data from both the Terra and Aqua satellites.
**Measurements Of Pollution In The Troposphere (MOPITT)**
MOPITT is a multi-channel thermal infrared (TIR) and near-infrared (NIR) instrument operating
on board the sun-synchronous polar-orbiting NASA Terra satellite. This study uses a version 9
(MOP03TM_9) gridded monthly product (Worden et al., 2010; Deeter et al., 2019). For more
details about the retrieval algorithm, validation, and uncertainties in MOPITT CO, refer to Deeter
et al. (2019).
**Atmospheric Infrared Sounder (AIRS)**



In addition to the MOPITT measurements, we utilized CO from the AIRS on the NASA Aqua
satellite, which provides CO at different vertical levels twice daily and has near-global coverage.
AIRS uses wavenumbers 2183–2200 cm−1 (4.58–4.5 µm) for retrieving CO (McMillan et al.,
2005). The V9 level 3 CO product, available at 1∘ × 1∘ resolution at various pressure levels, was
utilized in the present study. AIRS sensitivity to CO is broad and optimal in the mid-troposphere
between approximately 300 and 600 hPa (Warner et al., 2007, 2013; AIRS project, 2019). CO
retrievals exhibit a 6%–10 % bias between 900 and 300 hPa with a root mean square error of 8%–
12 % (McMillan et al., 2011). In addition to CO, we also utilized ozone, skin temperature, and
outgoing longwave radiation (OLR) data from the AIRS satellite.
**Ozone Monitoring Instrument (OMI)/Microwave Limb Sounder (MLS)**
We utilized the OMI/MLS dataset of global tropospheric column ozone (TCO) concentrations,
covering the period from 2005 to 2023, obtained from the Ozone Monitoring Instrument (OMI)
and the Microwave Limb Sounder (MLS) (Ziemke et al., 2006). The total ozone column from OMI
is derived using the Total Ozone Mapping Spectrometer (TOMS) version 8 algorithm. MLS
measures vertical ozone profiles above the upper troposphere via limb scans ahead of the Aura
satellite. TCO is then determined by subtracting MLS's stratospheric ozone measurements from
OMI's total column ozone, after calibration adjustments between the two instruments via the
convective-cloud differential method (Ziemke et al., 2006). The OMI/MLS product provides
monthly mean TCO data between 60°S and 60°N at a 1° × 1° resolution, starting from October
2004. This dataset has been extensively used to analyze global tropospheric ozone patterns
(Ziemke et al., 2019; Cooper et al., 2010) and long-term trends (Gaudel et al., 2018; Lu et al.,

154   2019).

**MERRA-2 reanalysis products**
We also utilized monthly mean geopotential height, wind vectors (zonal and meridional wind
speed), total column black carbon, organic carbon, and particulate matter from the Modern-Era
Retrospective Analysis for Research and Applications, version 2 (MERRA-2). MERRA-2 is the
latest atmospheric reanalysis data produced by the NASA Global Modeling and Assimilation
Office (GMAO; Gelaro et al., 2017). The horizontal resolution of the MERRA-2 reanalysis is 0.5°
× 0.625°.



**Soil Moisture**
Monthly mean soil moisture content (10 - 40 cm underground) from the Global Land Data
Assimilation System (GLDAS)_NOAH025_M v2.1 is utilized. The data can be downloaded from
https://hydro1.gesdisc.eosdis.nasa.gov/data/GLDAS/GLDAS_NOAH025_M.2.1/ (last accessed:
June 05, 2025).
**Precipitation**
The Global Precipitation Climatology Project (GPCP) Version 3.2 Satellite-Gauge (SG)
Combined Precipitation Data Set was used during the study period. The data is available for
download from https://measures.gesdisc.eosdis.nasa.gov/data/GPCP/GPCPMON.3.2/ (last
accessed June 5, 2025).
**2.1.2 Methodology**
The anomalies in the various parameters for April 2023 were estimated by subtracting the
background long-term mean for April (2003-2022) from the value for April 2023.
The magnitude of the AOD/CO enhancement in April 2023 above the long-term background was
determined by comparing the average of April 2003-2022. We obtained the percentage change in
AOD/CO relative to the respective background using Equation 1:
$$\text{Relative change in percentage} = \left(\frac{x_{i-\bar{x}}}{\bar{x}}\right) \times 100 \qquad \text{(Eq. 1)}$$
where $x_i$ represents the monthly mean of April in 2023, and $\bar{x}$ is the long-term mean of April
calculated using the data from 2003 to 2022.
**3. Results and Discussion**
**3.1 Record-breaking AOD and CO anomalies over SCS in April 2023**
Aerosol optical depth (AOD) is a standard measure used to estimate aerosol loading and is a key
parameter in calculating radiative effects. We utilize AOD data from MODIS instruments on the
Aqua and Terra satellites from 2003 to 2023. **Sup. Figures 2a** and **2b** show the long-term (2003-
2022) average AOD for April and the monthly mean AOD for April 2023. Time series of average



monthly AOD values over northern PSEA (17-23 N, 99-106 E) and SCS (109-119 E, 11-18 N)
from 2003 to 2023 are shown in **Sup—Figures 2c** and **2d**.

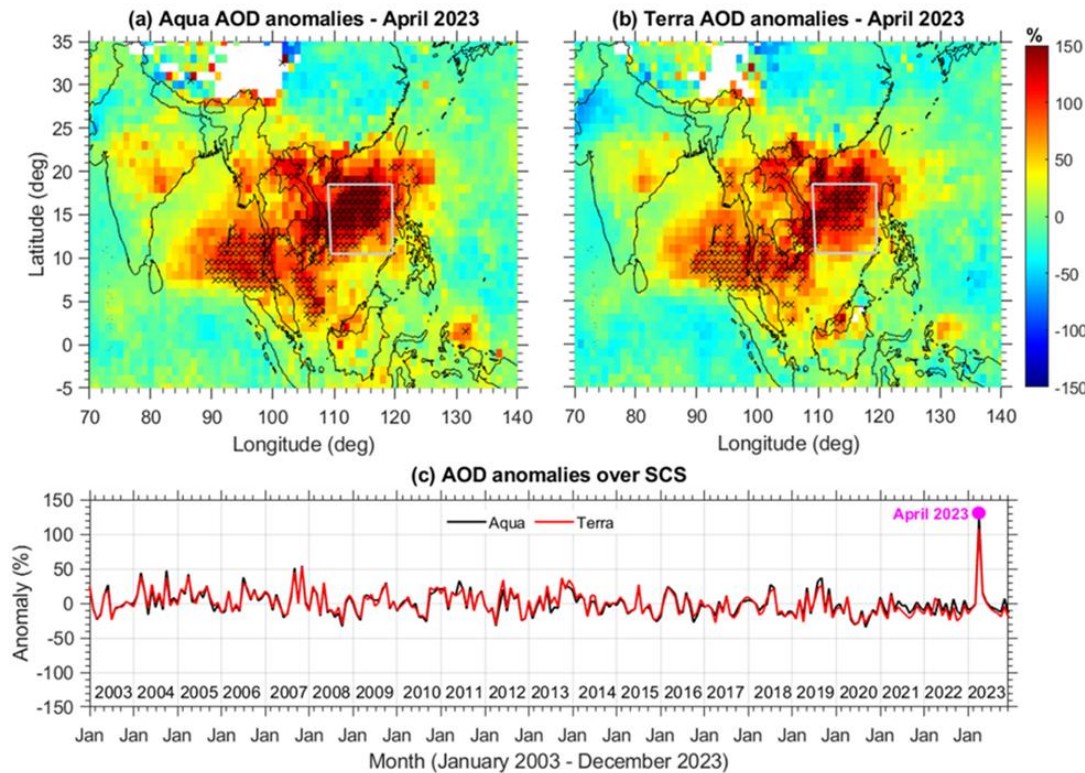


**Figure 2.** Spatial distribution of the change (%) in April 2023 Aerosol Optical Depth (AOD) values
compared with the inter-annual April average (2003-2022). (a) AOD anomalies are obtained from
the MODIS Aqua and (b) from the MODIS Terra satellite. The black hatches indicate that the
anomalies exceed 4σ standard deviations of the long-term mean. (c) Time series of area-averaged
AOD anomalies expressed in percentage change over the South China Sea (SCS) domain from the
Aqua (black line) and Terra (red line) satellites. The most significant enhancement was in SCS,
where the April AOD anomalies fell more than 4σ standard deviations.
The AOD distribution in April over two decades indicates high aerosol loading from northern Laos
to coastal South China (15-25 N, 100-120 E). In April 2023, extreme AOD values extended from
PSEA to South China and SCS, with the highest center between northern Laos and the SCS.
Record-breaking AOD levels were observed for the area averaged over the SCS in April 2023,
showing a nominal increase in northern PSEA (**Sup. Fig. 2c-d**). The highest AOD value for
northern PSEA in April 2023 correlates with record AOD over the SCS. To assess the magnitude
of the increase, we estimated the percentage change in AOD by comparing April 2023 with the



long-term average for April from 2003 to 2022. **Figures 2a** and **2b** depict the spatial extent of
AOD anomalies expressed as percentage changes from MODIS Aqua and Terra. A surprising and
widespread enhancement, with an increase of over 150% in most of the SCS and the southern Bay
of Bengal (BoB), was evident in April 2023, and the increased anomalies exceeded approximately
four standard deviation units. The area-averaged AOD anomalies (%) over the SCS domain from
Aqua (black line) and Terra (red line) satellites show that the increase in April 2023 was a record
high compared to MODIS data from 2003 to 2023, highlighting the extremity of AOD
enhancement in that month. Satellite observations were further corroborated by ground-based in
situ measurements from the AErosol RObotic NETwork (AERONET). The only operational
AERONET remote station downwind of PSEA biomass burning, with over a decade of continuous
AOD measurements (**Sup. Fig. 3a**), within the SCS region is located on Dongsha Island (also
known as Pratas Island, 20.70°N, 116.73°E; 5 m a.s.l.). Analysis of the monthly mean AOD data
from Dongsha Island indicates that April 2023 recorded the highest AOD value in the entire
observational period from January 2009 to December 2023 (**Sup. Fig. 3c**).
We further investigated CO changes in April 2023 across the study region using MOPITT
and AIRS satellite measurements, which provide over two decades of continuous CO data. CO is
a crucial trace gas due to its role as a tropospheric pollutant, atmospheric transport tracer, and
involvement in tropospheric chemistry. We analyzed CO data at 700 and 500 hPa from both
satellites between 2003 and 2023. The 500 hPa level is the most sensitive altitude for CO
measurements (Buchholz et al., 2021). The observed CO anomalies from the two satellites are
shown in **Figure 3**, highlighting significantly elevated CO levels in April 2023 over the SCS, with
increases up to $3\sigma$ standard deviations compared to the climatology from 2003 to 2022. Both
instruments reveal distinct spatial anomalies, with MOPITT displaying more concentrated CO
anomalies than AIRS. However, both show positive CO anomalies at both levels, indicating a
significant increase in CO in April 2023. It is worth noting that the spatial distribution of CO
anomalies aligns closely with AOD anomalies (**Fig. 2**). The area-averaged anomalies of AOD and
500 hPa CO over the SCS from 2003 to 2023 revealed a significant positive correlation of
approximately 0.65 (**Sup. Figure 4**). This strong correlation in April (2003–2023) suggests that
long-range pollution transport likely drives AOD variability in this region during April. The bubble
chart illustrates the severity of April 2023 compared to other years over the SCS (**Sup. Figure 4**).
CO production from incomplete combustion indicates that elevated levels far from traffic or



industrial sources suggest biomass burning and wildfire emissions. This implies smoke transport
from surrounding regions to the SCS and the BoB, near significant BB hotspots, including the MC
and PSEA. **Sup. Figure 1c-1d** shows annual AOD fluctuations, with peaks in April over PSEA
and September over MC. The MC's fire season is from August to October, while PSEA experiences
a BB season from January to April, peaking in March. This strongly suggested that the high AOD
levels in April 2023 over the SCS were linked to BB activities in PSEA.

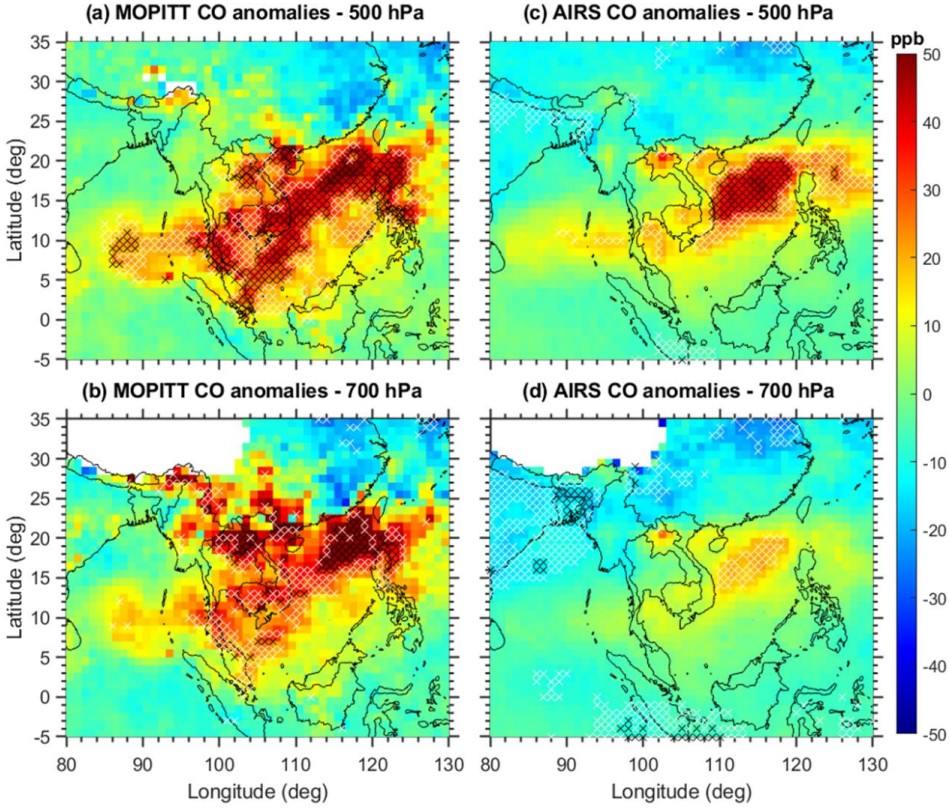


**Figure 3**. Spatial distribution of carbon monoxide anomalies in April 2023 at (a) 500 hPa and (b)
700 hPa obtained from MOPITT satellite measurements. Subplots (c) and (d) are identical to
subplots (a) and (b), but the results are obtained from AIRS satellite measurements. The anomalies
compared to the long-term mean of April from 2003 to 2022. The black and white hatches indicate
that the anomalies are more significant than 3σ and 2σ standard deviations, respectively.




**3.2 Biomass Burning activity over PSEA in April 2023**

We further examined BB activity over PSEA in April 2023, using MODIS fire counts and fire radiative power (FRP) as proxies for BB activities and wildfires. It is noted that there were a total of 21198 fire counts and a total FRP of 2407283 (MW) throughout PSEA (**Sup. Table 1**). Fire counts and FRP varied significantly between countries, with Laos reporting the highest number of fire counts (11877) and FRP (1530000 MW). Notably, Laos accounted for 56% of the total fire counts and 63% of the FRP for PSEA, establishing it as a hotspot for BB activity in April 2023. **Figures 4a** and **b** show the spatial distribution of fire counts and the related FRP over PSEA in April 2023. Persistent and more intense fires were observed over northern Laos and Myanmar, with the most intense fires occurring north of Laos. The number of fires in Laos in April 2023 was the highest recorded in the past 20 years (**Fig. 4c**). It should be noted that Laos is characterized by around 60% of its land cover types being forests (**Sup. Fig. 5**). Most of this forest is situated in northern Laos, where most fires occurred in April 2023. This suggests that forest fires in Laos were primarily responsible for the majority of fires in 2023. Although the FRP in 2023 was not at its peak, it was still among the highest BB activities, following 2016 and 2003. However, the nighttime fires and corresponding FRP demonstrate that the 2023 BB activity was the highest in the entire MODIS data record and exceptionally intense in terms of FRP (**Figs. 4e-4f**). This demonstrated the intensity of the fires in April 2023 compared to the last 20 years. The extreme nighttime fire activity highlights changes in fire behavior and environmental or human factors that favored intense nighttime burning in April 2023. We further examined area-averaged fires and FRP over northern Laos (17-23ºN), indicating that the highest fires and FRP were recorded in April 2023 in MODIS data from January 2003 to December 2023 (**Sup. Fig. 6**). Furthermore, the MODIS estimated monthly burned area product (MCD64A1) reveals a total area of 1.08 million hectares burned in Laos in April 2023, the highest monthly value in the available data for that dataset (2002–2023; **Sup. Fig. 7**). The spatial distribution of the MODIS burned area (**Sup. Fig. 8a**) shows that the most significant area affected by fires in 2023 was located in northern Laos, which closely aligns with the total number of fires and the FRP illustrated in **Figure 4**. This raises the question: What caused the anomalous fire activity in Laos in April 2023? We examined various meteorological and dynamic conditions in April 2023 to address this.

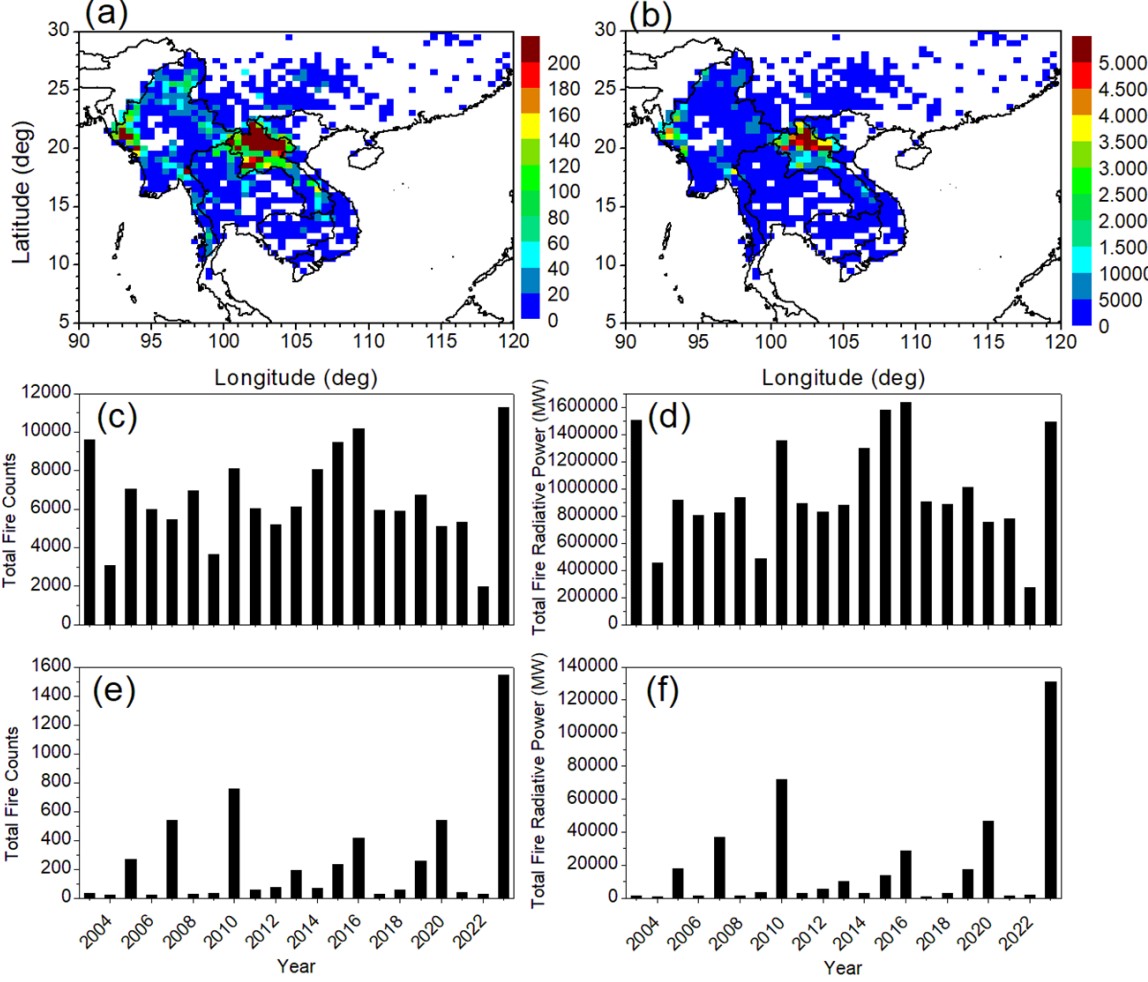

**Figure 4**. Spatial distribution (0.25° × 0.25°) of MODIS (a) fire counts and (b) total fire radiative power (FRP) in April 2023. A notable increase in fire activity over northern Laos is observed. (c) Inter-annual (2003 to 2023) monthly fire counts (day and night), and (d) the total FRP for April over Laos. (e) Inter-annual monthly nighttime fire counts and (f) the total FRP for nighttime fire counts over Laos in April. Fires with a confidence level of more than 80% are considered for the present analysis.

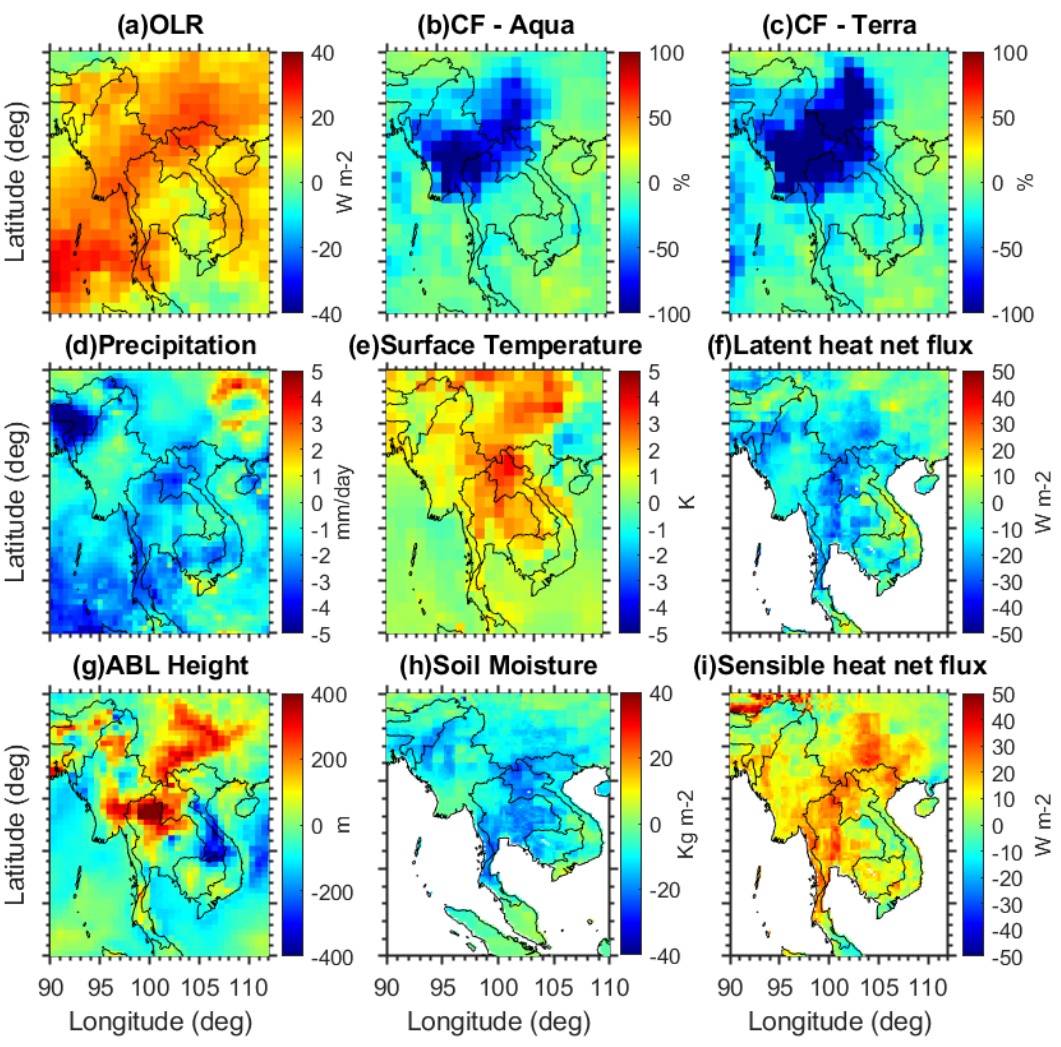

**Figure 5**. Surface and atmosphere conditions in April 2023. April anomalies in 2023 compared to the 2003-2022 climatological period for (a) Outgoing Longwave radiation (OLR), (b) cloud fraction (CF) from Aqua, (c) cloud fraction from Terra, (d) precipitation, (e) Surface Temperature, (f) surface latent heat flux, (g) Atmospheric Boundary Layer (ABL) Height, (h) soil moisture (10 - 40 cm underground), and (i) surface sensible heat flux. OLR and surface temperatures are obtained from AIRS satellite measurements. CF data from MODIS Aqua and Terra. ABL height obtained from MERRA-2 reanalysis. Soil moisture, surface latent heat, and sensible heat flux are obtained from the GLDAS Noah Land Surface Model L4 monthly 0.25 x 0.25 degree V2.1. Precipitation data is obtained from the Global Precipitation Climatology Project (GPCP) Version 3.2.

In April 2023, Outgoing Longwave Radiation (OLR) anomalies reflected decreased convective activity over PSEA, resulting in reduced precipitation, higher temperatures, and low





soil moisture (SM), as shown in **Fig. 5**. Precipitation, temperature, and SM anomalies correlate
with enhanced MODIS fire counts and FRP over northern Laos (**Fig. 4**). Long-term SM anomalies
in northern Laos reached record lows in April 2023, the lowest in two decades (**Sup. Figs. 9a and**
**b**). We further examined the evolution of SM anomalies during 2021-2023, which indicates
maximum positive values in March 2022 and maximum negative anomalies in April 2023,
signifying prolonged drought from winter 2022 to April 2023 (**Sup. Fig. 9c**). Interestingly, the
record-low SM anomalies occurred during the transition period from La Niña to El Niño. Under
dry conditions, increased sensible heat flux warms the near-surface atmosphere, resulting in a
positive land-atmosphere feedback (Alexander 2011). To explore how record SM anomalies in
April 2023 affected land-atmospheric coupling, we analyzed surface heat flux changes over PSEA,
revealing decreased surface latent heat flux and increased sensible heat flux. The negative latent
heat flux anomalies indicated limited evapotranspiration due to dry soil conditions. A deeper
Atmospheric Boundary Layer (ABL) height (>400 m increase) was observed over northern PSEA,
particularly in Laos, aligning with other anomalies in April 2023. Negative SM anomalies favor a
positive geopotential height anomaly in upper levels, maintaining local high pressure and
promoting surface warming (Fischer et al., 2007; Dong et al., 2023). The northern PSEA
experienced a high-pressure anticyclone in April 2023 (**Fig. 6a**), which decreased cloud cover,
increased solar radiation and surface temperature, and reduced precipitation. MODIS cloud
fraction (CF) anomalies exhibited extreme negative values, particularly in the northern PSEA,
decreasing by over 100% compared to the 2003-2022 average, closely aligning with the high-
pressure anticyclone depicted in **Figure 6a**. Reduced cloud cover and drier soil will increase heat,
landscape flammability, and wildfire potential. It is concluded that record-breaking negative SM
anomalies under a deeper, drier, and warmer ABL, coupled with increased temperatures, low
precipitation, and anomalous low cloud cover associated with the upper tropospheric high-pressure
system, contributed to record-breaking BB and wildfires over Laos in April 2023.
**3.3 Dynamical and large-scale circulations in April 2023**
Previous research shows that in spring, smoke aerosols (BC and OC) and trace gases are
transported from PSEA to downstream areas like southern SC, Taiwan, and the northwestern
Pacific via free tropospheric westerlies (Wai et al., 2008; Lin et al., 2009; Lin et al., 2013; Yen et
al., 2013; Chuang et al., 2014; Ou-Yang et al., 2014; Lin et al., 2017; Pani et al., 2019).



Remarkably, in April 2023, PSEA BB smoke aerosols and trace gases were transported to the far
southern regions of the SCS and the BoB, marking a significant departure from the usual transport
pathway to downwind Taiwan and the northwest Pacific, respectively.

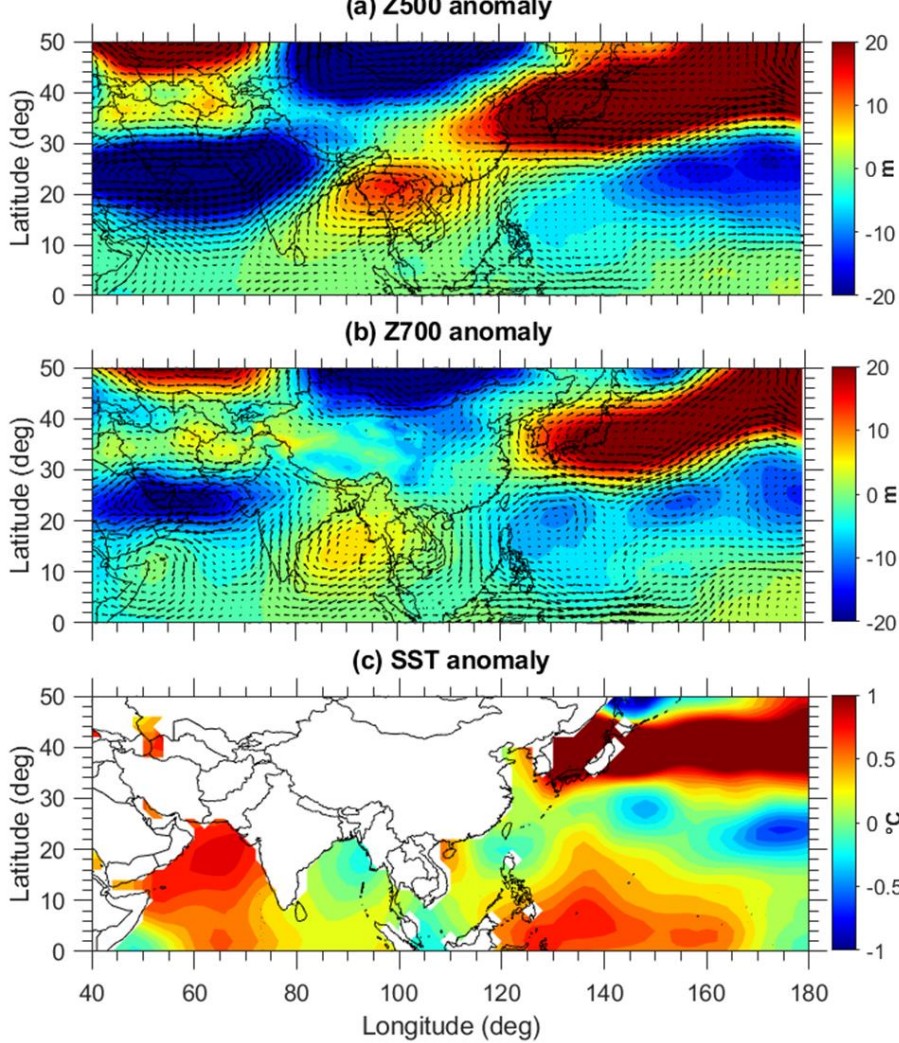


**Figure 6.** Spatial distribution of (a) 500 hPa geopotential height (Z500), (b) 700 hPa geopotential
height (Z700), and (c) Sea Surface Temperature (SST) anomalies in April 2023. The anomalies
are calculated by subtracting the monthly mean of April 2023 from the April climatology for the
period from 1991 to 2020. The wind anomalies for the respective pressure levels are overlaid in
Z500 and Z700 anomalies. The geopotential height and wind data are from the MERRA-2
reanalysis, while SST data are from the NOAA Extended Reconstructed SST V5.





Here, background dynamics and large-scale circulations are crucial for transporting smoke
aerosols over longer distances from the sources (Ravindra Babu et al., 2023; Huang et al., 2024).
To provide context for the April 2023 event, we briefly analyzed the large-scale circulation
patterns responsible for the unprecedented aerosol loading. Our focus was on the geopotential
height and winds, both zonal and meridional, at 700 and 500 hPa levels. The geopotential height
observed at these levels (Z700 and Z500) in April 2023 contrasts with the background climatology
of April (1991-2020) (**Sup. Figs. 10 and 11**). Specifically, at 700 hPa (Z700), a high-pressure
system over the Indian region shifted eastward, reaching the PSEA. In comparison, at 500 hPa
(Z500), the western Pacific anticyclone shifted westward to sit directly above the PSEA. To get a
clearer picture, we further obtained the anomaly in Z700 and Z500 in April 2023 by comparing
the long-term mean of 1991-2020. The observed Z700 and Z500 anomaly composites are
illustrated in **Figure 6a and b**. There is a prominent anomalous anticyclone over northern PSEA,
centered roughly at 20°N, 100°E (**Fig. 6a**). Additionally, a significant anomalous low-level
cyclone is present over the western North Pacific (WNP) around the Philippines, with an
anomalous cyclone forming upstream and downstream of the anticyclone over the PSEA, creating
a zonal low-high-low (L-H-L) pattern. This arrangement might suggest the movement of a Rossby
wave train (Hu et al., 2024). Concurrently, a strong anticyclone anomaly was situated over the
northern Pacific Ocean, just above the western Pacific cyclone anomaly. It is strongly indicated
from **Figures 5** and **6** that the upper- and lower-level anomalous anticyclones significantly caused
cloudless skies, reduced precipitation, and elevated surface temperatures in the PSEA. These
favorable conditions, occurring over drier soil, led to extreme BB and wildfires in Laos, which
released significant quantities of aerosols and trace gases into the atmosphere.
We hypothesize that these systems, including an anomalous WNP cyclone, a BoB
anticyclone at 700 hPa, and a high-pressure anticyclone over PSEA at 500 hPa, substantially
influenced the background circulations in the PSEA and its surroundings. This interaction likely
contributed to the unusual BB smoke transport from northern PSEA to the SCS and the BoB in
April 2023, as corroborated by the meridional wind anomalies in **Sup**. **Figs. 12b and c**, which
indicate unusual northerly winds over the SCS. The northerly wind anomalies inhibited smoke
transport from northern PSEA to the SCS, resulting in unprecedented aerosol loading over the SCS
and the BoB. Additionally, zonal wind anomalies showed typical background westerlies
supplanted by easterlies over the North Pacific near Japan, due to an anomalous high-pressure



system in April 2023 (**Sup. Figs. 12a and b**). This high-pressure system weakened the westerlies
to its south, obstructing the usual smoke transport from PSEA to the downwind northwestern
Pacific. This is further supported by AERONET observations from Lulin Atmospheric
Background Station (LABS, 23°28′N, 120°52′E, 2,862 m; Sheu et al., 2010), located downwind
of the PSEA smoke (**Sup. Fig. 12a**). The AOD data revealed no notable rise in AOD at Lulin in
April 2023 compared to previous years, similar to what was observed at Dongsha Island (**Sup.**
**Fig. 13c**). In conclusion, the unusual circulation from the BoB anticyclone and the WNP cyclone
transported PSEA smoke into the SCS, resulting in unprecedented aerosol loading in April 2023.
Exploring the causes and dynamics behind the anti-cyclonic and cyclonic circulations over the
BoB and WNP in April 2023 is intriguing; however, it falls outside the scope of this study.
Preliminary analysis of sea surface temperature (SST) anomalies in April 2023 revealed a distinct
and spatially coherent pattern across the Pacific Ocean (**Fig. 6c**). Notably, positive SST anomalies
were observed over the western Pacific warm pool near the equatorial region, while negative
anomalies appeared over the central to eastern equatorial Pacific. In addition, strong positive SST
anomalies were present over the mid-latitude North Pacific. This tri-polar SST structure is known
to influence large-scale atmospheric circulation patterns. According to the Matsuno–Gill
framework, enhanced warming in the tropical western Pacific can induce anomalous cyclonic
circulation over the WNP (Gill, 1980; Zeng and Sun, 2022). Concurrently, regional SST anomalies
over the Indian Ocean exhibited positive values in the Arabian Sea and negative anomalies in the
BoB. These SST anomalies corresponded closely with 700 hPa geopotential height (Z700)
anomalies, which were positive over the BoB and negative over the Arabian Sea. The spatial
alignment of SST and Z700 anomalies suggests that the observed SST anomaly configuration in
April 2023 likely exerted a substantial influence on tropospheric circulation over both the Indian
and Pacific Oceans.
**3.4 Impact on Tropospheric Ozone**
It is well known that BB smoke can emit aerosols and various gaseous compounds,
including nitrogen oxides ($NO_x$), CO, methane ($CH_4$), and multiple volatile organic compounds
(VOCs). Once emitted, BB smoke undergoes chemical transformations in the atmosphere, altering
the mix of compounds and generating secondary pollutants such as ozone ($O_3$) and secondary
organic aerosol (Jaffe and Wigder, 2012; Ogino et al., 2022). BB emissions from the PSEA have
a significant impact on air quality and weather in both source and downwind regions. In previous




sections, we demonstrated the record-breaking increase in AOD, along with the unusual
enhancement in tropospheric CO (at 700 and 500 hPa) over SCS due to PSEA biomass burning. It
is known that the presence of CO is one of the factors that control the abundance of tropospheric
ozone, a short-lived pollutant and climate forcer (Liu et al., 1999; Chan et al., 2003; Ou-Yang et
al., 2012; Yadav et al., 2017; Liao et al., 2021).

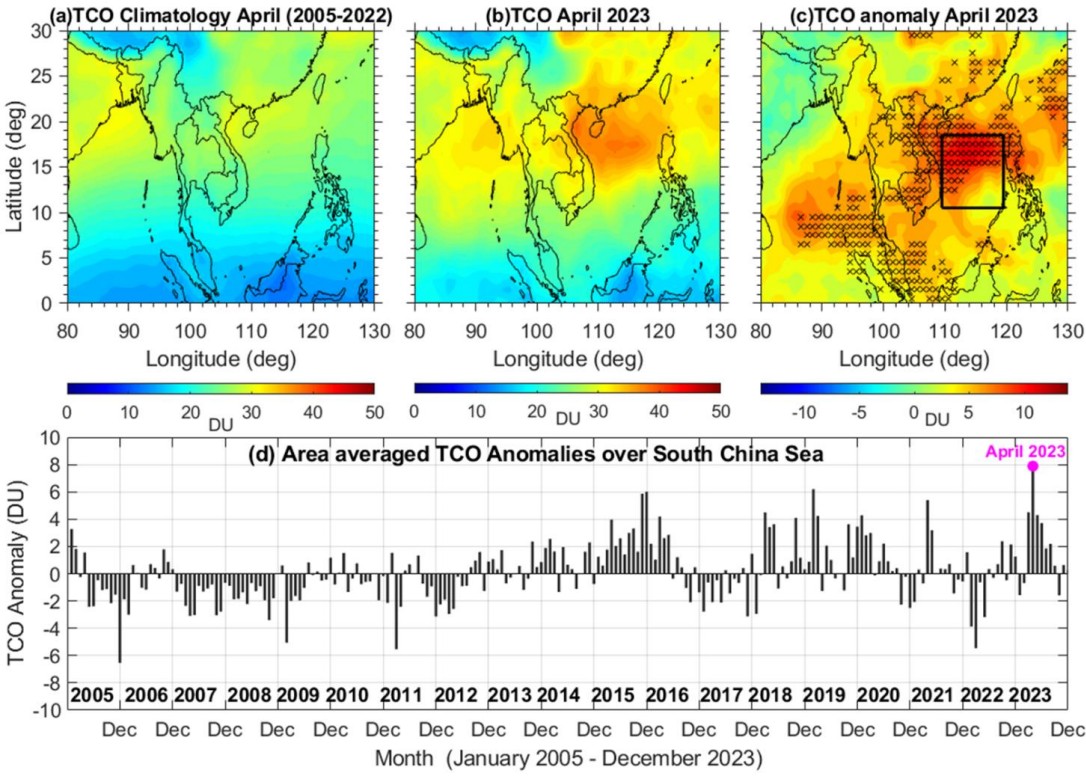


**Figure 7**. Spatial distribution of tropospheric column ozone (TCO) (surface - 300 hPa ozone column) concentrations in (a) April (2005-2022), (b) April 2023. (c). The observed spatial distribution of TCO anomaly in April 2023. Anomalies based on removing the 2005-2022 April mean. (d) The area-averaged TCO anomalies observed over SCS between January 2005 and December 2023—anomalies based on removing the long-term mean from 2005 to 2022. The highest increase in the TCO for the SCS region is recorded in April 2023 during the OMI/MLS data period. The black hatches in sub-plot (c) indicate that the anomalies are more significant than 3σ standard deviations, respectively.

Here, we investigated how this record-breaking pollution event may have influenced tropospheric
ozone levels in April 2023. We analyzed long-term tropospheric ozone column (TOC) data
(surface to 300 hPa) from the combined Aura Ozone Monitoring Instrument and Microwave Limb





Sounder satellite ozone measurements (OMI/MLS) from 2005 to 2023 (Ziemke et al., 2006; 2019).
**Figures 7a** and **7b** illustrate the spatial distribution of the long-term April mean and the April 2023
TOC over the study area. Additionally, **Figure 7c** illustrates the observed anomaly in April 2023
in comparison to the long-term average. We also include area-averaged TOC anomalies over the
SCS spanning the entire OMI/MLS dataset. The OMI/MLS TOC anomaly indicates significantly
elevated ozone levels over the SCS and nearby regions during April 2023. The observed anomalies
are statistically significant, being three standard deviations above the long-term mean of total
column ozone (TOC). Furthermore, the monthly mean anomalies, averaged over the SCS
throughout the entire OMI/MLS data period, reveal the highest increase in TOC, approximately 8
Dobson Units (DU), in April 2023. These exceptional TCO increases from the OMI/MLS data are
further supported by the AIRS satellite $O_3$ measurements and the downwind ozonesonde
measurements at Hong Kong, which are presented in **Sup. Figure 14**, respectively. We utilized
long-term ozone measurements from the AIRS satellite from 2003 to 2023. Our analysis of 700
and 500 hPa levels reveals a substantial $O_3$ increase over SCS and nearby areas, about 20 ppb,
exceeding two standard deviations of the long-term mean, corroborated by downwind Hong Kong
ozonesonde measurements (**Sup. Figs. 14c and d**). The ozone profile peaks at altitudes of 3 to 4
km, with anomalies exceeding 30 ppb in the 3 to 5 km region, correlating exactly with the
CALIPSO vertical aerosol enhancement. This illustrates the exceptional augmentation of TOC
over the SCS in April 2023, comparable to the AOD increase observed in the MODIS data. Such
findings are corroborated by the corresponding increases in CO levels recorded by MOPITT and
AIRS at 700 hPa and 500 hPa, respectively, indicating a substantial influence of BB plumes
originating from the PSEA region in 2023. Overall, the present analysis concludes that the April
2023 event had a significant impact on air quality over PSEA and its surrounding areas.
**4. Summary and Conclusions**
In April 2023, we observed an unprecedented increase in aerosol loading over the South
China Sea (SCS), which had not been observed in the past two decades of the MODIS period,
spanning from 2003 to 2023. Satellite observations revealed a 150% rise in aerosol optical depth
(MODIS), alongside 50% increases in carbon monoxide (MOPITT) at 700 and 500 hPa over SCS.
This study primarily focused on analyzing the drivers, physical and dynamical mechanisms behind
the record-breaking aerosol loading over the SCS in April 2023. Our findings indicate that extreme



biomass burning (BB) activity over northern PSEA was the primary source of the record-breaking
aerosols in April 2023.

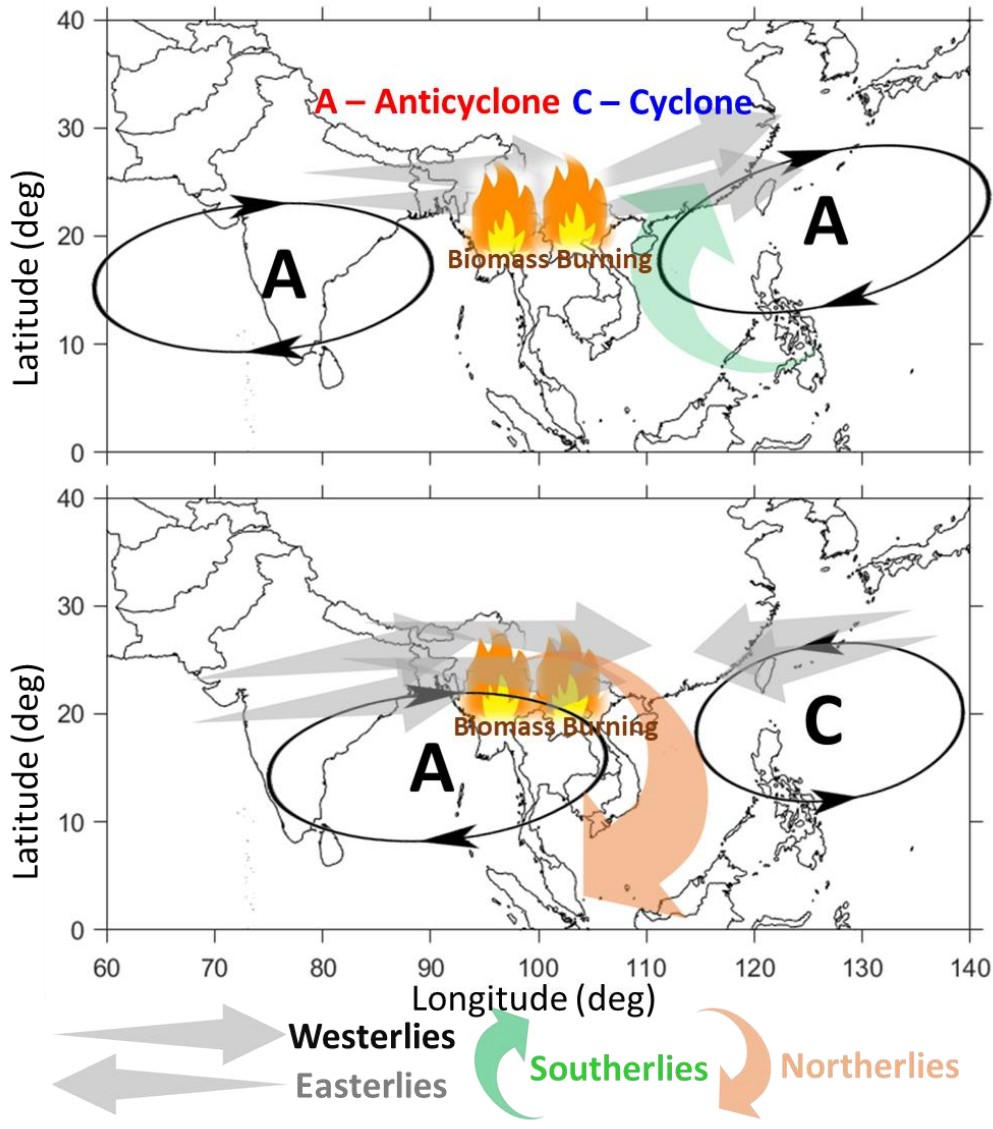


**Figure 8.** The schematic diagram illustrates the physical mechanisms responsible for the record-
breaking aerosol loading over the South China Sea in April 2023. The top panel displays the long-
term mean state in April, whereas the bottom panel shows the April 2023 mean state of the large-
scale dynamical and circulatory systems. A denotes the presence of an anticyclone anomaly, and
C represents the presence of a cyclone anomaly. The horizontal arrows indicate subtropical free-



tropospheric westerlies and easterlies. The green arrow indicates southerlies, and northerlies are shown by the brown arrow, respectively. Two anticyclone systems were present in climatology over the western Pacific and the Indian Ocean. Southwesterly and southerly winds in the free troposphere dominate the SCS. Free tropospheric westerlies transport smoke into the downwind areas of Taiwan and the western North Pacific. In April 2023, the western North Pacific anticyclone transitioned into an anomalous cyclone over the western North Pacific. The Indian anticyclone system further shifted eastward around the PSEA. Unusual northerly winds replaced the southerly winds due to a cyclone anomaly over the western North Pacific and an expanded Indian anticyclone. Additionally, in April 2023, easterlies around Taiwan and above hindered downwind transport to the northwestern Pacific.

An analysis of various meteorological and atmospheric factors reveals that the PSEA region has experienced unusual weather patterns, creating conditions conducive to BB and wildfires. Key contributors include extremely low soil moisture, higher surface temperatures, lower precipitation levels, and an upper tropospheric high-pressure anticyclone. These factors increase the likelihood of severe fire events, especially in Laos and Myanmar. Particularly, Laos became one of the hotspot regions for extreme BB activity in April 2023. Among all the countries in PSEA, Laos alone contributed approximately 56% of the total fire activity over PSEA, followed by Myanmar at around 33%. Under prolonged dry conditions, BB activity over Laos in April 2023 was higher than in the past two decades. The largest area burned, 1.08 million hectares, in a single month (2002-2023), occurred in April 2023. Additionally, unusually large-scale atmospheric circulations significantly spread smoke, trace gases, and pollutants to downwind regions from the source. Our analysis of large-scale circulations associated with dynamical changes illustrates the mechanism behind the April 2023 event, as schematically shown in **Figure 8**. In climatology, two anticyclone systems were situated over the WNP and the Indian Ocean. The SCS experiences predominantly southwesterly and southerly winds in the free troposphere. Westerly winds in the free troposphere generally transport BB smoke from PSEA to downwind areas of Taiwan and the WNP. In April 2023, the anticyclone over the WNP transformed into an unusual cyclone. Meanwhile, the Indian anticyclone shifted eastward over the BoB and near the PSEA. Due to a cyclone anomaly in the WNP and a persistent anticyclone in the BoB, northerly winds replaced the southerly winds in the free troposphere over the SCS. Additionally, in April 2023, easterlies near Taiwan obstructed downwind transport towards the northwestern Pacific. Overall, it is concluded that the regime shifted from southerlies to northerlies over the SCS due to the combined impact of the extended BoB anticyclone and the WNP cyclone, causing BB smoke transport from the PSEA to the SCS.



The present findings would benefit regional monitoring and a better understanding of the
transboundary pollution over the PSEA.
Interestingly, PSEA is linked to an extreme heatwave in April 2023, with record-high
temperatures (Zachariah et al., 2024; Lyu et al., 2024). Studies have attributed this heatwave to
climate change (Zachariah et al., 2024), as well as to the strengthening of high pressure from
tropical waves, moisture deficiency, and strong land-atmosphere coupling (Lyu et al., 2024). Our
results further suggest a plausible role for BB-associated aerosols and greenhouse gases in the
April 2023 heatwave. What role does heat trapping play in increasing greenhouse gases resulting
from record-breaking BB activity? What is the impact of increased BB aerosols? Further research
is needed to understand the exceptional conditions in PSEA and its surrounding regions, including
BB-associated greenhouse gas emissions (GHGs) and aerosol anomalies. Additionally, smoke
aerosols impact surface and atmospheric radiation budgets, affecting regional weather and climate.
Future work will focus on the radiative energy balance and weather changes resulting from the
April 2023 aerosol increase.
**Data availability**
MODIS data available from https://modis.gsfc.nasa.gov/data/dataprod/mod08.php. The AIRS and
MOPITT CO data can be downloaded from https://disc.gsfc.nasa.gov/datasets/AIRS3STM_7.0
(AIRS project, 2019) and https://asdc.larc.nasa.gov/project/MOPITT/MOP02J_8 (NASA, 2023a).
MERRA-2 data are available online through the NASA Goddard Earth Sciences Data Information
Services Center (GES DISC; https://disc.gsfc.nasa.gov; NASA, 2023b). The MODIS fire products
can be downloaded from https://firms.modaps.eosdis.nasa.gov/active_fire/ (NASA, 2023c). The
OMI/MLS tropospheric column ozone data can be obtained from https://acd-
ext.gsfc.nasa.gov/Data_services/cloud_slice/ (last access 01 July 2025).
**Author Contributions**
**Saginela Ravindra Babu:** Conceptualization, Data curation, Formal analysis, Investigation,
Software, Validation, Visualization, Writing – original draft preparation, Writing – review and
editing; **Neng-Huei Lin**: Conceptualization, Investigation, Funding Acquisition, Supervision,
Resources, Writing – review and editing.



**Competing interests**

The authors declare no competing interests.

**Acknowledgements**

We acknowledge the National Science and Technology Council of Taiwan for supporting the research. The authors thank NASA and NOAA for providing MOPITT, MODIS, and AIRS satellite data. We thank NASA's Global Monitoring and Assimilation Office (GMAO) for providing the Modern-Era Retrospective analysis for Research and Applications, Version 2 (MERRA-2) data.

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
