# Peer review of "Sea in April 2023"

_EGUsphere, 2025_

## Author Comment (AC1)

**Response to Referee #1**

 General Comment

 The manuscript addresses the record-breaking aerosol loading over the South China Sea (SCS) in
 April 2023, attributed to biomass burning (BB) over the northern Indochina Peninsula. While the
 topic is of regional and global importance, the study suffers from several critical issues. The
 methodology is overly simplistic, the novelty is limited, the logical flow is confusing, and key
 presentation elements (maps, data classification, figures) do not meet the standards of a top-tier
 journal. In its current form, the manuscript reads more like a descriptive case report rather than an
 in-depth scientific analysis. Substantial revision is needed before it can be considered for
 publication.

 Reply: We thank the reviewer for the critical review of our original manuscript, which has helped
 us revise it for the better. Our point-by-point responses to the review comments are provided
 below. The authors sincerely thank the reviewer for dedicating the time and effort to evaluating
 our manuscript and for providing thoughtful, constructive feedback. We also appreciate the
 reviewer's recognition of the significance of our study. Their valuable comments have greatly
 contributed to enhancing the quality and clarity of our work.

 All figures have been revised and improved in accordance with the reviewers' suggestions to
 enhance clarity and precision.

 Major Comments

 **Scientific Significance and Novelty**

 Biomass burning over Indochina and its long-range transport to the SCS is a well-documented and
 recurring phenomenon (e.g., Lin et al., 2013; Reid et al., 2013). The manuscript merely shows that
 April 2023 recorded the highest anomalies in AOD/CO/ozone in the past two decades. Without
 deeper analysis of what makes 2023 fundamentally different (e.g., unique transport pathways,
 distinct chemical mechanisms, significant health/climate impacts), the work risks being a replication of prior studies with little added value. The authors need to explicitly demonstrate the **novelty** and scientific importance of this case beyond being "the largest on record."

Reply: The authors fully recognize the well-established link between biomass burning in Indochina (PSEA) and its transport to the South China Sea (SCS), Taiwan, and the western North Pacific, as shown in earlier research (Lin et al., 2013; Reid et al., 2013). The authors are well aware of the

PSEA BB activity, its transport mechanisms, and its effects on regional weather and climate.

However, the reviewer may have misunderstood the study's primary focus and its significant findings. While previous research indicates that most transported smoke stays north of about

17.5°N over the SCS, our analysis shows that **during April 2023, the smoke was unusually**

**transported much farther south, reaching the southern parts of the SCS and even extending**

**toward the southern Bay of Bengal (BoB).** This departure from typical patterns is significant.

To illustrate this, we present the average April AOD distribution for 2003–2022 in Supplementary

Figure 2 (now it is Figure 3 in the revised manuscript), which shows higher AOD levels across approximately 17.5°N-25°N, from northern Indochina to Taiwan. Conversely, the SCS region outlined in a black box generally exhibits very low AOD. The April 2023 event is notable for its intensity and spatial coverage. The AOD anomalies obtained are 4 times the long-term mean over most of the SCS and the southern BoB region in April 2023. We believe our study highlights a rare, previously unreported transport event, not simply reproducing prior work, with potential consequences for regional air quality and climate. We will emphasize this aspect further in the revised manuscript to clearly showcase the novelty and importance of the April 2023 case.

**Mismatch Between Analysis and Conclusions**

The conclusions claim clear attribution to Laos fires and anomalous circulation systems. However, the analysis is largely descriptive, relying on anomaly maps and percentage changes. The causal chain (fire activity → transport anomalies → AOD/CO increases → ozone formation) is not rigorously substantiated. For example, CO–AOD correlation (~0.65) only suggests coincidence, not causality. Ozone enhancement is attributed to BB emissions without distinguishing between primary transport and secondary chemistry. The authors should either strengthen the causal evidence (e.g., trajectory modeling, chemical transport simulations, Rossby wave diagnostics) or tone down the conclusions.

For example, CO–AOD correlation (~0.65) only suggests coincidence, not causality.

**Reply:** We appreciate the reviewer's valuable comments. In the revised manuscript, we have
strengthened the analysis by incorporating HYSPLIT back-trajectory analysis, vertical aerosol
profiles from CALIPSO images, and vertical changes of black carbon (BC) and organic carbon
(OC) from MERRA-2 reanalysis. In addition, we have included an analysis of formic acid
(HCOOH) to better illustrate the role of secondary photochemical processes in ozone formation
over the South China Sea (SCS). These additions have been reflected in revised conclusions, with
careful consideration of the evidentiary limits of the present study.

Regarding the CO–AOD relationship, we note that the manuscript mistakenly reported a
correlation coefficient of 0.65; this is in fact the coefficient of determination ($R^2$). The
corresponding correlation coefficient is R = 0.81, indicating a statistically robust association
between CO and AOD over the SCS. While we agree that correlation alone does not prove
causality, the strong CO–AOD relationship in this remote, marine region, largely free of local
anthropogenic sources, supports the interpretation of long-range transport of combustion-related
aerosols. Elevated CO, a tracer of incomplete combustion, observed far from urban and industrial
sources, is consistent with biomass burning (BB) influence. Seasonal consistency further supports
this interpretation: annual AOD maxima over Peninsular Southeast Asia (PSEA) in March–April
and over the Maritime Continent in September coincide with their respective peak fire seasons. To
strengthen source attribution, we analyzed daily HYSPLIT back trajectories, which show that air
masses arriving over the SCS during April 2023 predominantly originated from northern PSEA,
consistent with active BB regions during this period (Figure R1). While this trajectory analysis
does not constitute a complete transport attribution framework, it provides dynamical support for
BB influence.

We agree with the reviewer that ozone enhancement should not be attributed solely to BB
emissions without distinguishing between primary transport and secondary photochemical
production. In the revised manuscript, we explicitly clarify this distinction. Ozone anomalies over
the SCS coincide with elevated CO and AOD; however, quantifying ozone production would
require chemical transport modeling, which is beyond the scope of this study. Nevertheless,
additional observational evidence supports secondary chemical processing within transported BB

plumes. Infrared Atmospheric Sounding Interferometer (IASI) MetOp satellite observations show a ~100% enhancement in formic acid (HCOOH) over the SCS in April 2023 (Figure R2). HCOOH

is a well-established secondary oxidation product of VOCs emitted by biomass burning. The simultaneous enhancements of CO (>50 ppb), AOD (~150%), and HCOOH indicate that BB

plumes underwent substantial photochemical aging during transport. Therefore, the observed ozone enhancement is interpreted as primarily influenced by secondary ozone formation within transported BB plumes, rather than by direct ozone transport alone.

[Figure]

Figure R1. (a) Daily 72-h NOAA HYSPLIT backward trajectories ending at 12:00 UTC at a
representative location (15°N, 115°E) over the South China Sea at 3 km altitude, overlaid on
MODIS fire counts for April 2023. (b) Same as (a), but overlaid on the monthly mean MODIS
aerosol optical depth (AOD) for April 2023.

[Figure]

Figure R2. The Infrared Atmospheric Sounding Interferometer (IASI) METOP satellite observed total column HCOOH (a) April long-term mean (2008-2022), (b) April 2023, (c) the percentage change in HCOOH in April 2023 compared to the long-term mean (2008-2022). (d) inter-annual variability of HCOOH in April, and (e) the observed percentage change anomaly in HCOOH over the South China Sea.

**Methodology Too Simplistic**

The methodology is limited to anomaly calculations relative to the 2003–2022 climatology and σ-thresholds. No advanced statistical diagnostics (EOF, regression, composite analysis) or modeling tools (WRF-Chem, GEOS-Chem, HYSPLIT) are applied. For a high-impact journal, such purely descriptive methods are insufficient. More mechanistic or quantitative approaches are expected to justify publication.

**Reply**: We appreciate the reviewer's comment. Calculating anomalies relative to a long-term climatological period (2003–2022 in our case) is a well-established and widely accepted method in atmospheric and climate sciences (Avery et al., 2017; Hirsch and Koren, 2021; Hedelius et al., 2021; Rieger et al., 2021; Stone et al., 2025; Prasanth et al., 2025). Comparing these anomalies with the corresponding monthly standard deviations provides a quantitative measure of their extremity. In our study, we clearly state that the observed AOD anomaly in April 2023 exceeded the long-term mean by more than four standard deviations, indicating an exceptionally event. It is also noted that similarly focused studies have successfully employed such methods. For example, Hirsch and Koren (2021), in their *Science* article ("Record-breaking aerosol levels explained by smoke injection into the stratosphere"), used fundamental anomaly analysis to identify record-high AOD levels resulting from the Australian wildfires (See the attached figure below). This highlights that the suitability of methodology depends on the study objective, and complex statistical techniques are not always required for publication in high-impact journals.

[Figure]

**Fig. 2 The SH anomaly for January 2020 (spatial resolution of 5° by 5°).**

(**A**) Interannual (2003 to 2019) monthly average AOD values for January. (**B**) Monthly AOD values for January 2020. A notable increase in the AOD values over the SH is observed. (**C**) The change in January 2020 AOD values compared with the interannual January average (expressed in standard deviation units). (source: Hirsch and Koren, 2021, *Science*.)

However, we fully agree with the reviewer that approaches such as EOF analysis, regression, and chemical transport modeling (e.g., WRF-Chem, GEOS-Chem) are highly valuable for exploring underlying mechanisms and causal relationships. Following the reviewer's helpful suggestion, we have included HYSPLIT backward trajectory analysis in the revised manuscript to provide additional evidence for long-range transport from biomass-burning regions (see Figure R1).

We also examined AOD variability across the study region using EOF analysis; the results are shown in the attached figure (Figure R3) for your reference. An EOF analysis was applied to the observed monthly mean AOD time series in the study region (90-120E, 5-25N) to determine the dominant modes of variability over the period. The spatial distribution and temporal amplitude are negative, resulting in a positive value. A higher negative value indicates higher AOD. Figure 2(a)

shows higher AOD in the northern PSEA and the coastal area of southern China. The result of EOF1

* PC1 (multiplication) is the same.

[Figure]

Figure R3. (a) The spatial distribution, and (b) its corresponding time-varying amplitude for the
vector EOF analysis mode 1 of the April AOD in SCS during 2003 to 2023.

We agree that chemical transport models such as WRF-Chem and GEOS-Chem could provide further insight into the chemical and physical processes involved. However, incorporating such models is beyond the scope of this observational and event-focused study. We will clearly state this limitation in the revised manuscript and consider it a priority for future research.

**Logical Flow and Structure**

The introduction devotes excessive space to global wildfire events (Canada, Hawaii, Mediterranean), which dilutes the focus on the SCS case.

Reply: Thank you for your insightful comment. The discussion of global wildfire events (e.g., in Canada, Hawaii, and the Mediterranean) was included in the introduction to highlight the unusually active and widespread nature of wildfires during the study period, placing the South China Sea (SCS) event within a broader global context. However, we understand that this may have diluted the focus on the SCS case. In response, we have revised the introduction to briefly summarize the global activity while more clearly emphasizing the relevance and distinctiveness of the SCS aerosol episode, ensuring that the central focus of the study remains clear.

The Results and Discussion section frequently shifts between AOD, CO, fire counts, meteorology, circulation, and ozone, without a clear hierarchical structure. This leads to a confusing narrative.

The manuscript would benefit from a re-organization: Phenomenon confirmation → Source attribution → Circulation mechanisms → Chemical/ozone impacts → Implications.

Reply: Thanks for the voluble suggestion. We have reorganized the results and discussion section in the revised manuscript as suggested by the reviewer.

**Data Classification and Transparency**

- Satellite products (MODIS, MOPITT, AIRS, OMI/MLS), reanalysis datasets (MERRA-2, GLDAS, GPCP), and in-situ measurements (AERONET, ozonesondes) are all mixed together in one section.
- It is difficult for the reader to distinguish between direct observations, model-assimilated reanalysis, and ground truth data.
- The Data and Methodology section should be reorganized into clear categories: (1) Satellite remote sensing, (2) Reanalysis/model products, (3) Ground-based observations.

Reply: Thanks for the voluble suggestion. We have modified the Data and Methodology section in the revised manuscript as suggested. We also included a table describing the data used in the present study.

Table R1. Details of various data products used in the present study.

| Data | Resolution | Source |
|---|---|---|
| Aerosol Optical Depth (AOD) | $1° \times 1°$ | Aqua and Terra satellite/MODIS |
| Carbon Monoxide (CO) | $1° \times 1°$ | MOPITT and AIRS |
| Tropospheric Column Ozone (TCO) | $1° \times 1°$ | OMI/MLS |
| Burned Area (BA) | 500 m | Aqua and Terra satellite/MODIS |
| MODIS Collection 6.1 Fire Anomalies | | combined Terra and Aqua satellite/MODIS |
| Wind and Geopotential Height | $0.5° \times 0.625°$ | MERRA reanalysis |

**Use of Supplementary Figures**

Key evidence (e.g., climatological AOD distributions, long-term time series) is presented only in
Supplementary Figures. Essential results should be in the main text, with Supplementary reserved
for additional details or robustness checks. As written, the paper is not self-contained.

Reply: We appreciate the reviewer's valuable comment. We agree that the climatological AOD
distributions and long-term time series provide essential context. As suggested by the reviewer, we
have moved the key figures showing the climatological AOD distributions and the long-term AOD
time series from the Supplementary Materials to the main text (now Figures 2 and 3, respectively).

**Map Presentation and Political Sensitivity**

Several figures show solid boundary lines in regions with disputed territories (e.g., South China
Sea). International journals require disputed boundaries to be indicated with dashed lines and/or with a neutral disclaimer in the captions. The authors must revise all maps accordingly to comply
with cartographic and editorial standards.

Reply: We appreciate the reviewer's careful observation and constructive comment. Following the
recommendation, we have revised all maps to comply with editorial standards.

**Lack of Impact Assessment**

The study stops at describing anomalies. There is no evaluation of downstream consequences
(e.g., impacts on regional air quality, radiative forcing, health risks). Without such discussion, the
significance of the findings remains limited.

Reply: Thank you for your comment. The primary objective of this study is to identify the drivers
and underlying physical mechanisms responsible for the record-breaking aerosol loading over the
South China Sea. To provide some insight into potential impacts, we focused specifically on
associated ozone changes in the present paper. A more comprehensive evaluation of the effects on
radiative forcing, atmospheric processes, and air quality is beyond the scope of this study. It will be
addressed in a follow-up study.

Minor Comments
Figures are overcrowded, with small fonts and inconsistent styles (gradient colors vs. hatching).
Improve readability and adopt a uniform design.
Reply: We have taken utmost care in the figures in the revised manuscript.
Figure 8 schematic is overly simplistic compared to the complexity of earlier figures; it should more
clearly contrast climatological vs. 2023 circulation states.
Reply:
Reference formatting is inconsistent; some entries are incomplete or lack DOI.
Reply: Corrected in the revised manuscript
The writing style is verbose. The introduction should be shortened and sharpened to highlight the
scientific problem.
Reply: Corrected in the revised manuscript
The format are not clearly uniform between 1∘ × 1∘ in L116 and 0.25° in L124. The font format of
L124-125 is different from other context.

Reply: Corrected in the revised manuscript

L140, why the skin temperature is used in this work?

Reply: It is a typo mistake. We used the surface temperature from the AIRS satellite. We have corrected this typo in the revised manuscript.

L187, L198, add ° for the logitude and latitude.

Reply: Corrected in the revised manuscript

L185, Sup. Figures, L188, Sup—Figures, P201, Sup. Fig. and so on, keep the same citaiton style, refer to the papers in the top journals.

Reply: Corrected in the revised manuscript

L 241, the maps are not correct, as we know, there are still undecided boarders between China and India, the author should clearly state them in the maps.

Reply: Corrected in the revised manuscript

References

Hirsch, E. and Koren, I.: Record-breaking aerosol levels explained by smoke injection into the stratosphere, Science, 371, 1269–1274, https://doi.org/10.1126/science.abe1415, 2021.

Avery, M., Davis, S., Rosenlof, K. et al. Large anomalies in lower stratospheric water vapour and ice during the 2015–2016 El Niño. Nature Geosci 10, 405–409 (2017). https://doi.org/10.1038/ngeo2961

Hedelius, J. K., Toon, G. C., Buchholz, R. R., Iraci, L. T., Podolske, J. R., Roehl, C. M., Wennberg, P. O., Worden, H. M., and Wunch, D.: Regional and urban column CO trends and anomalies as observed by MOPITT over 16 years, J. Geophys. Res.-Atmos., 126, e2020JD033967, https://doi.org/10.1029/2020JD033967, 2021.

Rieger, L. A., Randel, W. J., Bourassa, A. E., and Solomon, S.: Stratospheric Temperature and Ozone Anomalies Associated With the 2020 Australian New Year Fires, Geophys. Res. Lett., 48, e2021GL095898, https://doi.org/10.1029/2021GL095898, 2021.

Stone, K., Solomon, S., Yu, P., Murphy, D. M., Kinnison, D., and Guan, J.: Two-years of stratospheric chemistry perturbations from the 2019–2020 Australian wildfire smoke, Atmos. Chem. Phys., 25, 7683–7697, https://doi.org/10.5194/acp-25-7683-2025, 2025.

Prasanth, S., Anand, N. S., Sunilkumar, K., Jose, S., Arun, K., Satheesh, S. K., and Moorthy, K. K.: Australian bushfire emissions result in enhanced polar stratospheric clouds, Atmos. Chem. Phys., 25, 7161–7186, https://doi.org/10.5194/acp-25-7161-2025, 2025.

We once again thank the reviewer for carefully reviewing the manuscript and for offering potential solutions that helped us significantly improve its content.

---

## Author Comment (AC2)

**Response to Referee #2**

This paper investigates the drivers of the record-breaking aerosol loading event over the South China Sea in April 2023, combining satellite observations, reanalysis data, and ground-based measurements. The study convincingly shows that large-scale biomass burning in northern Laos and Myanmar, combined with anomalous circulation patterns, caused unprecedented aerosol transport into the SCS. The integration of multiple datasets (MODIS, MOPITT, AIRS, OMI/MLS, and MERRA-2) strengthens the analysis, and the paper provides timely insights into extreme aerosol events under changing climate conditions. Overall, this work makes a valuable contribution to understanding regional transboundary pollution processes. I recommend minor revisions before acceptance.

Reply: We thank Reviewer #2 for the positive review and fair remarks, which have all been carefully implemented in the manuscript.

(1) The fire activity analysis over Laos is a highlight of the study. Since you mention that 2023 had the largest monthly burned area on record, adding a supplementary figure comparing 2023 with other extreme fire years (e.g., 2016, 2003) would strengthen the historical context.

Thank you for your valuable suggestion. We have already included the monthly burned area for Laos from January 2003 to December 2023 in Supplementary Figure 7. Additionally, Supplementary Figure 8 illustrates the interannual variability in April's burned area over the same period. These two figures clearly show that the largest monthly burned area during 2003–2023 was in April 2023, totaling 1.08 Mha.

Following the reviewer's suggestion, we further examined the MODIS burned area data for April 2003, 2016, and 2023 and present a comparison of their spatial distributions in the figure below (Figure R1). The results indicate that the spatial distributions of burned areas in 2016 and 2023 are generally similar, with the most extensive burning occurring in northern Laos compared with 2003. However, the burned area in 2023 was more concentrated and extensive in northern Laos than in 2016. We further included the overall burned-area variability over PSEA (combining all PSEA countries) from 2003 to 2023 in April as a supplement figure in the revised manuscript. We noticed that the burned area in April 2023 is the highest with 2.27 Mha during the 2003 to

2023 period for the PSEA region. We also included each country's Burned Area (BA) and the contribution percentage for the total BA in Table 1. For your reference, we have attached the interannual variability of BA for April over PSEA in Figure R2.

[Figure]

Figure R1. The spatial distribution of MODIS burned area (BA) in April 2003, 2016, and 2023.
(Source: https://firms.modaps.eosdis.nasa.gov/)

[Figure]

Figure R2. Interannual variability in Burned Area over Peninsula Southeast Asia in April from
2003 to 2023.

(2) The study identifies biomass burning in Laos as the major contributor to the April 2023 event.

Could the authors clarify whether other regional fire sources (e.g., Maritime Continent or southern

China) were quantitatively excluded, or whether their contributions are negligible compared to

Laos?

Reply: Thank you for the insightful comment. After a thorough investigation of the spatial distribution of burned areas and fire counts, we confirmed that the contributions from the Maritime

Continent and southern China were negligible for the extreme AOD observed in April 2023. As shown in Figure R3, the burned area was predominantly concentrated over the PSEA region, with little to no burning detected over the Maritime Continent or southern China. Therefore, these regions were excluded from our calculations.

[Figure]

Figure R3. Spatial distribution of the MODIS Global Burned Area Product in April 2023. (Source:
https://firms.modaps.eosdis.nasa.gov/

(3) The circulation analysis (anticyclone over Bay of Bengal and cyclone over WNP) is central to the conclusions. It would help if the authors could briefly discuss whether such anomalous circulation patterns are unique to 2023, or if similar circulation shifts have occurred in past years
without producing record-breaking aerosol loading.

Reply: We thank the reviewer for this insightful comment. In the revised manuscript, we examined
historical circulation patterns during other high-biomass-burning years over the study region. The
accompanying Figure R4 compares April 2023 with the high-BB composite, allowing an
assessment of whether the circulation and aerosol conditions in 2023 resemble those commonly
observed during severe biomass-burning periods. The comparison reveals notable differences
between 2023 and other high-BB years (Figure R4). In particular, April 2023 is characterized by
a pronounced anticyclonic high-pressure system over PSEA that is stronger and more spatially
coherent than in the high-BB composite. Correspondingly, AOD levels in 2023 are substantially
higher than those in the high-BB composite, indicating unusually intense aerosol loading. These
distinctions suggest that the circulation configuration in 2023 may have played a greater role in
aerosol accumulation and transport than in typical high-biomass-burning years.

[Figure]

Figure R4. Spatial distribution of MODIS aerosol optical depth (AOD) and MERRA-2 500-hPa geopotential height with wind vectors for April: (a) AOD composite for high biomass-burning years, (b) AOD for 2023, (c) 500-hPa geopotential height and wind vectors for high biomass-burning years, and (d) 500-hPa geopotential height and wind vectors for 2023.

(4) The study shows a strong correlation (r ~ 0.65) between AOD and CO anomalies. Could the authors expand on the physical interpretation? For example, does this imply biomass burning was the sole driver, or might secondary aerosol formation also have amplified AOD?

Reply: We thank the reviewer for the insightful comment. The authors want to clarify that the manuscript mistakenly reported a correlation coefficient of 0.65; in fact, this is the coefficient of determination ($R^2$). The corresponding correlation coefficient is R = 0.81, indicating a statistically robust association between CO and AOD over the South China Sea (SCS). The AOD enhancement over the South China Sea was predominantly transport-driven. That said, we do not exclude the role of secondary aerosol formation. However, the further investigation of vertical aerosol distribution from CALIPSO images and the MERRA-2 black carbon (BC) and organic carbon (OC), followed by the NOAA HYSPLIT back trajectories, clearly demonstrates that the PSEA BB is the primary factor for the record-breaking aerosol loading over SCS in April 2023.

(5) The discussion links record-low soil moisture in Laos with enhanced fire intensity. Would it be possible to show a supplementary time series of soil moisture anomalies alongside fire counts to more directly demonstrate this relationship?

Reply: We thank the reviewer for this helpful suggestion. In response, we have added a supplementary time series showing standardized fire anomalies alongside soil moisture anomalies over Laos (Figure R5). The analysis indicates that low soil moisture is significantly correlated with enhanced fire activity during April 2023, with 2023 exhibiting an extreme negative soil moisture anomaly concurrent with strong positive fire anomalies. A similar co-occurrence of anomalously low soil moisture and elevated fire activity is also evident in 2016, a year previously identified as having intensified burning. These results provide more direct observational support for the link between soil moisture deficits and enhanced fire intensity, and the new figure is included in the Supplementary Material.

[Figure]

Figure R5. Inter annual variability in (a) standardized fire anomalies, (b) soil moisture anomalies over Laos in April during 2003 to 2023.

(6) Figure 2-4 provides rich information, but it is quite dense. For readers who are not familiar with the dataset, more explanatory notes or simplified illustrations (such as highlighting Laos as a fire hotspot) can improve accessibility. Figure 2c can be changed in color to highlight the contrast between Aqua and Terra.

Reply: We thank the reviewer for this helpful suggestion. In the revised manuscript, we have replotted most of the figures to improve clarity and accessibility. Additional explanatory notes have been added where appropriate, and key features such as Laos as a major biomass-burning hotspot are now more clearly highlighted. In addition, Figure 2c has been revised with an updated color scheme to better distinguish and emphasize the contrast between Aqua and Terra MODIS observations.

(7) The schematic diagram in Figure 8 is excellent. Consider changing the color scheme of A/C cyclone and east-west wind to improve readability.

Reply: We thank the reviewer for the positive and encouraging comment. In the revised manuscript, we have further improved the schematic diagram in Figure 8 by adjusting the color scheme of the anticyclonic/cyclonic circulation and the east–west wind components to enhance visual contrast and readability. The revised figure is provided below for reference (Figure R6).

[Figure]

**Figure R6.** Schematic diagram illustrating the physical mechanisms driving the record-breaking
aerosol loading over the South China Sea in April 2023. A indicates an anticyclonic anomaly, and
C indicates a cyclonic anomaly.

(8) The description of datasets is clear, but it would be useful to briefly summarize in one table the different satellite/reanalysis products used, their spatial/temporal resolutions, and the key variables. This would make the methodology section more reader-friendly.

Reply: In response to the reviewer's comment, we have now included a table summarizing the datasets.

Table R1. Details of various data products used in the present study.

| Data | Resolution | Source |
|---|---|---|
| Aerosol Optical Depth (AOD) | $1° \times 1°$ | Aqua and Terra satellite/MODIS |
| Carbon Monoxide (CO) | $1° \times 1°$ | MOPITT and AIRS |
| Tropospheric Column Ozone (TCO) | $1° \times 1°$ | OMI/MLS |
| Burned Area (BA) | 500 m | Aqua and Terra satellite/MODIS |
| Fire Anomalies | $0.25° \times 0.25°$ | Aqua and Terra satellite/MODIS |
| Wind and Geopotential Height | $0.5° \times 0.625°$ | MERRA reanalysis |

(9) Since the paper emphasizes Southeast Asian fire climatology, it may be helpful to cite prior
works that have quantified the magnitude and variability of fire activity in this region, such as
Cohen (2014) and Cohen et al. (2017). Adding these references would provide a stronger
background for the discussion of extreme fire activity in 2023. (https://doi.org/10.1088/1748-
9326/9/11/114018; https://doi.org/10.5194/acp-17-721-2017)

Reply: We thank the reviewer for this helpful suggestion. We have added the recommended
references (Cohen, 2014; Cohen et al., 2017) to the revised manuscript and incorporated them into
the background and discussion sections. These studies provide essential context on the magnitude,
spatial distribution, and interannual variability of fire activity in Southeast Asia, and they strengthen the framing of the extreme biomass-burning conditions observed in 2023 relative to historical variability.

(10) When mentioning black carbon transport and associated trace gases, the authors may consider citing recent top-down studies on BC and CO emissions in Asia (e.g., Wang et al., 2021; Wang et al., 2025). These works would complement the current study by highlighting related emission and transport perspectives. (https://doi.org/10.1029/2021EF002167; https://doi.org/10.1038/s41612-025-00977-2)

Reply: We thank the reviewer for this helpful suggestion. In the revised manuscript, we have incorporated a discussion of changes in black carbon (BC) and organic carbon (OC) in April 2023 based on the MERRA-2 reanalysis. In addition, we have added the recommended top-down studies on BC and CO emissions and transport over Asia (Wang et al., 2021; Wang et al., 2025), which provide valuable complementary perspectives on emission strength and long-range transport. These references strengthen the broader context of our findings and help link the observed aerosol and trace-gas enhancements to regional emission and transport processes.

**We once again thank the reviewer for carefully reviewing the manuscript and for offering potential solutions that helped us significantly improve its content.**

---

## Author Comment (AC3)

**Response to Referee #3**

The manuscript presents a thorough analysis of an unprecedented aerosol loading event over the South China Sea (SCS) in April 2023, using multiple satellite datasets and reanalysis products. The authors convincingly identify biomass burning in Laos and Myanmar as the primary source, and they discuss the unusual circulation anomalies that directed smoke transport into the SCS. The study is timely, relevant, and potentially impactful, especially given the increasing frequency of climate–fire extremes. However, I believe the manuscript requires further development before it can be accepted. My major concerns relate to the quantification of uncertainties, the robustness of transport attribution, and the integration of climate drivers. I detail my comments below.

We highly appreciate the thoughtful and valuable suggestions from the reviewer, which will help us improve the quality of our manuscript. We have revised the manuscript with consideration of the reviewer's comments/suggestions.

The manuscript reports extreme anomalies in MODIS AOD ($>4\sigma$) and MOPITT/AIRS CO ($>3\sigma$), but little discussion is provided regarding retrieval errors, biases, or limitations. Please provide a clearer treatment of uncertainties, for example: known MODIS biases over ocean and land, vertical sensitivity limits in MOPITT CO, and representativeness of reanalysis aerosol products. A sensitivity analysis (e.g., comparison across Aqua vs. Terra MODIS, MOPITT vs. AIRS CO) would help quantify robustness.

Reply: We thank the reviewer for this critical comment. In the revised manuscript, we have expanded the discussion of uncertainties and limitations for the satellite and reanalysis products used in this study. We have also carried out comparisons between Aqua and Terra MODIS AOD, as well as MOPITT and AIRS CO at 500 hPa over the South China Sea (see attached Figure R1), to assess robustness.

For MODIS AOD, the estimated uncertainty is approximately $\pm0.05$ over ocean and $\pm0.15$ over land. The Collection 6.1 (C6.1) products used in this study have been shown to capture temporal variations effectively and agree closely with ground-based observations (Wei et al., 2019b). Validation against AErosol RObotic NETwork (AERONET) measurements demonstrates that the merged Dark Target and Deep Blue (DTB) products accurately capture aerosol variability at both regional and global scales (Sayer et al., 2014; Wei et al., 2019). Comparison of Terra and

Aqua MODIS AOD confirms consistent temporal patterns, with extreme anomalies exceeding 4σ

observed in both datasets.

For MOPITT CO, primary sources of uncertainty include vertical sensitivity limits and retrieval biases. The observed enhancements (>3σ) are consistent with independent AIRS CO

measurements, supporting the robustness of the reported anomalies. Although MOPITT's sensitivity decreases near the surface, combining both instruments' observations and applying quality filters mitigates this limitation.

We have added these discussions in the revised manuscript and highlighted that, despite known uncertainties, the extreme anomalies reported are robust across multiple datasets and instruments, including MODIS Aqua/Terra, MOPITT, AIRS, and reanalysis fields. Relevant validation studies are now explicitly cited (Sayer et al., 2014; Wei et al., 2019; Ziemke et al.,

2006).

[Figure]

Figure R1. (a) Comparison between MODIS Terra AOD and MODIS Aqua AOD, (b) comparison
between MOPITT and AIRS measured 500 hPa CO over the South China Sea during January 2003
to December 2023. (R is the correlation coefficient; N is the sample size; P is the significance
value).

Transport Attribution and Circulation Analysis

The explanation of northerly transport due to the Bay of Bengal anticyclone and western North

Pacific cyclone anomaly is plausible, but remains descriptive. I strongly recommend including trajectory or dispersion modeling (e.g., HYSPLIT, FLEXPART) to explicitly demonstrate that biomass burning plumes from Laos could reach the SCS. Alternatively, a composite analysis of circulation anomalies in other strong-fire years could be used to strengthen causality.

Reply: Thanks for the helpful suggestion. In the revised manuscript, we have included an analysis of the CALIPSO and MERRA-2 vertical aerosol distributions in April 2023. As aerosol optical depth (AOD) is a column-integrated measure, it does not provide information on the vertical distribution of aerosols. To overcome this limitation, we analyzed observations from the Cloud-

Aerosol Lidar and Infrared Pathfinder Satellite Observation (CALIPSO), which reveal pronounced enhancements of smoke aerosols over the South China Sea (SCS). Elevated smoke layers were also observed over the southern Bay of Bengal (BoB) in April 2023, predominantly within the mid-troposphere at altitudes of approximately 3–5 km. Consistent with these lidar observations,

MERRA-2 reanalysis data indicate substantial increases in aerosol mass concentrations in 2023, with black carbon (BC) increasing by ~250% and organic carbon (OC) by ~350%. The most pronounced enhancements occur between 700 and 600 hPa, closely matching the altitude range identified by CALIPSO. The concurrence of satellite and reanalysis evidence points to a severe pollution episode in April 2023 over and near the SCS, characterized by elevated aerosol layers indicative of long-range transported biomass-burning smoke. To examine the transport mechanism, we have further analyzed HYSPLIT back trajectories for April 2023. We have run daily HYSPLIT back trajectories at random (15N-115E, 3000 m), and the resulting trajectories are shown in the following Figure R2. It is clear that air masses arriving over the SCS during April

2023 predominantly originated from the northern PSEA, consistent with the active BB regions observed during this period.

[Figure]

Figure R2. Daily 72-h NOAA HYSPLIT backward trajectories ending at 12:00 UTC at a representative location (15°N, 115°E) over the South China Sea at 3 km altitude in April 2023.

Following the reviewer's suggestion, we analyze the large-scale circulation and aerosol loading in other high-biomass-burning (BB) years over Peninsular Southeast Asia (PSEA). High-BB years are objectively identified by calculating standardized fire anomalies from total MODIS fire counts over PSEA, with years exceeding a 0.5 threshold classified as high-BB (Figure R7). Using this criterion, composite fields of MODIS aerosol optical depth (AOD), 500-hPa geopotential height, and wind vectors are constructed to represent typical circulation and aerosol patterns associated with enhanced biomass-burning activity. The accompanying figure compares April 2023 with the high-BB composite, allowing an assessment of whether the circulation and aerosol conditions in 2023 resemble those commonly observed during severe biomass-burning periods. The comparison reveals notable differences between 2023 and other high-BB years (Figure R3). In particular, April 2023 is characterized by a pronounced anticyclonic high-pressure system over PSEA that is stronger and more spatially coherent than in the high-BB composite. Correspondingly, AOD levels in 2023 are substantially higher than those in the high-BB

composite, indicating unusually intense aerosol loading. These distinctions suggest that the
circulation configuration in 2023 may have played a greater role in aerosol accumulation and
transport than in typical high-biomass-burning years.

[Figure]

Figure R3. Spatial distribution of MODIS aerosol optical depth (AOD) and MERRA-2 500-hPa
geopotential height with wind vectors for April: (a) AOD composite for high biomass-burning
years, (b) AOD for 2023, (c) 500-hPa geopotential height and wind vectors for high biomass-
burning years, and (d) 500-hPa geopotential height and wind vectors for 2023.

**Link to Large-Scale Climate Drivers**

The manuscript notes the La Niña–El Niño transition and a tri-polar SST anomaly structure but does not fully connect these anomalies to the extreme biomass burning and circulation changes. Please expand the discussion to show whether such SST/ENSO anomalies have historically coincided with enhanced PSEA burning or altered circulation patterns. This would greatly strengthen the broader climate relevance of the study.

Reply: Thank you for this valuable suggestion. We agree that establishing a more explicit linkage between sea surface temperature (SST)/ENSO anomalies and regional fire and circulation responses will enhance the broader climate relevance of the study. The interannual variability of biomass-burning (BB) activity over Indochina has been closely tied to the El Niño–Southern Oscillation (ENSO), as reported in previous studies (Yin, 2020; Zhu et al., 2021; Zheng et al., 2023). ENSO is a dominant driver of interannual BB variability across South and Southeast Asia. During El Niño events, prolonged drought and suppressed precipitation intensify fire activity over northern Indochina, particularly in spring (Zhu et al., 2021). Zheng et al. (2023) further showed that fire occurrences increase substantially during El Niño years, coinciding with more fire-prone meteorological conditions compared to La Niña years. This asymmetry reflects stronger correlations between fire weather and the ENSO index during El Niño phases, associated with positive low-level geopotential height anomalies and reduced water vapor transport over Southeast Asia (March–May), both of which favor enhanced burning.

However, the record-breaking aerosol event in April 2023 occurred during the La Niña–El Niño transition, following an unusual triple-dip La Niña. This transitional state appears distinct from previously documented ENSO–fire relationships and may have contributed to atypical circulation and moisture anomalies. In the revised manuscript, we have expanded the discussion to highlight these connections and compare the 2023 transition pattern with historical ENSO phases, thereby emphasizing the broader climatic context of the observed extreme biomass burning.

Additionally, we constructed composites of MODIS AOD and 500-hPa wind vectors for El Niño and La Niña years during 2003–2022 (Figure R4). The results reveal an apparent increase in AOD over northern PSEA and the coastal regions of South China during El Niño years compared to La Niña years. The associated circulation patterns also differ, with El Niño years characterized by a stronger anticyclonic system over PSEA extending from the Bay of Bengal, consistent with enhanced aerosol accumulation in the region. These results support the interpretation that ENSO-related circulation anomalies strongly modulate regional aerosol loading and fire activity.

[Figure]

Figure R4. Composite fields of MODIS aerosol optical depth (AOD) and 500-hPa wind vectors
for April: (a) El Niño years, (b) La Niña years, and (c) the difference between El Niño and La Niña
composites.

**Minor Comments**

**Figures and Visualization**

Several figures (e.g., Figs. 2, 3, 5, 6) are visually dense with overlapping hatching and color contours. Please simplify or separate key results, and ensure legends are large and consistent.

Reply: We thank the reviewer for this suggestion. In the revised manuscript, we have improved the clarity and visual presentation of all figures.

**Terminology Consistency**

The text alternates between "TCO" and "TOC" for tropospheric ozone. Please standardize terminology throughout.

Reply: Corrected in the revised manuscript.

**Ground-Based Validation**

AERONET data from Dongsha Island and Lulin are mentioned but not analyzed in detail. I suggest including explicit time series plots and quantitative comparisons with satellite AOD to reinforce credibility.

Reply: We thank the reviewer for this suggestion. In the revised manuscript, we have added explicit details, including time-series plots and quantitative evaluations, to strengthen the credibility of the satellite observations. Specifically, we now provide direct comparisons between AERONET AOD and MODIS AOD at these sites, highlighting both seasonal variability and absolute agreement. Monthly AERONET AOD time series at Dongsha Island and Lulin are also included to provide context (Figure R5 and R6). These additions allow a more comprehensive assessment of the satellite-derived AOD and reinforce the reliability of our aerosol analysis.

[Figure]

Figure R5. (a) Comparison between AERONET AOD and MODIS Terra AOD, (b) AERONET AOD and MODIS Aqua AOD over Dongsha Island during January 2009 to December 2023. (R is the correlation coefficient; N is the sample size; P is the significance value)

[Figure]

Figure R6. (a) Comparison between AERONET AOD and MODIS Terra AOD, (b) AERONET AOD and MODIS Aqua AOD over LABS during January 2006 to December 2023. (R is the correlation coefficient; N is the sample size; P is the significance value)

Literature Context

The manuscript could benefit from more thorough discussion of prior SCS and Southeast Asian biomass burning studies (e.g., 7-SEAS campaigns, Lin et al. 2013; Reid et al. 2013). This would help contextualize the novelty of the April 2023 event.

Reply: We appreciate the reviewer's suggestion. Following the advice, we have discussed further
more about the previous 7-SEAS studies in the revised manuscript.

**Language and Style**
Some sentences are repetitive (e.g., emphasis on Laos' share of BB activity) and could be
streamlined. Please also ensure consistent reference to "Supplementary Figures" rather than "Sup.
Figures."

Reply: We thank the reviewer for this comment. We have carefully revised the manuscript to
streamline repetitive sentences, particularly those emphasizing Laos' contribution to biomass-
burning activity, and to improve clarity and readability. Additionally, we have standardized all
references to supplementary material, using "Supplementary Figures" consistently throughout the
text.

**Outlook / Future Work**
The conclusions briefly mention aerosol–radiation interactions and links to heatwaves. I encourage
a more explicit outlook section, highlighting next steps such as quantifying radiative forcing or
simulating impacts with chemistry–climate models.

Reply: We thank the reviewer for this valuable suggestion. In the revised manuscript, we have
expanded the outlook/future work section to provide a more explicit discussion of potential next
steps. Specifically, we highlight opportunities to quantify aerosol–radiation interactions and
estimate the associated radiative forcing, as well as to investigate the regional climate impacts,
including heatwaves, using chemistry–climate or Earth system model simulations. These
directions will help build on the present study by linking observed extreme biomass burning and
aerosol enhancements to broader climate and atmospheric consequences.

[Figure]

Figure R7. Inter-annual variability in (a) total fire counts, (b) the standardized fire anomalies over Peninsula Southeast Asia (PSEA) from 2003 to 2023.

We once again thank the reviewer for carefully reviewing the manuscript and for offering potential solutions that significantly improved its content.